# A fat-tissue sensor couples growth to oxygen availability by remotely controlling insulin secretion

Michael J. Texada [1], Anne F. Jørgensen [1,2], Christian F. Christensen[1], Takashi Koyama [1], Alina Malita[1], Daniel K. Smith[1], Dylan F. M. Marple[1], E. Thomas Danielsen [1], Sine K. Petersen[1], Jakob L. Hansen[2], Kenneth A. Halberg [1] & Kim F. Rewitz [1]

Organisms adapt their metabolism and growth to the availability of nutrients and oxygen, which are essential for development, yet the mechanisms by which this adaptation occurs are not fully understood. Here we describe an RNAi-based body-size screen in *Drosophila* to identify such mechanisms. Among the strongest hits is the fibroblast growth factor receptor homolog *breathless* necessary for proper development of the tracheal airway system. *Breathless* deficiency results in tissue hypoxia, sensed primarily in this context by the fat tissue through HIF-1a prolyl hydroxylase (Hph). The fat relays its hypoxic status through release of one or more HIF-1a-dependent humoral factors that inhibit insulin secretion from the brain, thereby restricting systemic growth. Independently of HIF-1a, Hph is also required for nutrient-dependent Target-of-rapamycin (Tor) activation. Our findings show that the fat tissue acts as the primary sensor of nutrient and oxygen levels, directing adaptation of organismal metabolism and growth to environmental conditions.

[1] Department of Biology, University of Copenhagen, 2100 Copenhagen, Denmark. [2] Cardiovascular Research, Department number 5377, Novo Nordisk A/S, Novo Nordisk Park 1, 2760 Måløv, Denmark. Correspondence and requests for materials should be addressed to K.F.R. (email: Kim.Rewitz@bio.ku.dk)

Multicellular organisms must adapt their metabolism and growth to limitations imposed by their environment. Genetic and physiological pathways such as growth factor systems regulate these adaptations, and these in turn are influenced by internal and external cues. Among these are nutrient availability, which promotes growth through the insulin/insulin-like growth factor (I/IGF) pathways[1,2], and oxygen sufficiency, which is also essential for the growth and development of animals[3,4]. The signaling pathways that regulate these fundamental aspects of development are evolutionarily ancient and well conserved.

I/IGF is the major systemic nutrient-dependent growth regulator and acts through conserved pathways to promote cellular growth[5,6]. Many connections between nutritional cues and systemic insulinergic regulation have been elucidated in the fruit fly *Drosophila melanogaster*. Several *Drosophila* insulin-like peptides (DILPs or simply insulin below), primarily DILP2, -3, and -5, are released into circulation from the insulin-producing cells (IPCs) of the brain[7]. Despite their different location, these cells are functionally homologous with mammalian pancreatic β-cells[8]. The DILPs signal through a single receptor (InR) to regulate both metabolism and growth. DILP expression and release are regulated in part by nutritional information relayed through the fat body, an organ analogous to vertebrate adipose and liver tissues with nutrient storage, metabolic, and endocrine functions. This tissue secretes insulinotropic (Unpaired-2, functionally analogous to the mammalian adipokine Leptin[9]; the peptide hormones CCHamide-2 (CCHa-2)[10], FIT[11], and Growth-Blocking Peptides (GPBs) 1 and 2[12,13]; the Activin ligand Dawdle (Daw)[14,15]; and the protein Stunted (Sun)[16]) and insulinostatic (the tumor necrosis factor-α homolog Eiger (Egr)[17]) factors, many of these in response to nutrient-dependent activity of the Target-of-rapamycin (Tor) pathway. Thus, this tissue is a central nexus for nutritional signals that mediate adaptation to nutritional deprivation.

Organisms require oxygen in addition to nutrients for growth and development and have therefore developed oxygen-sensing and adaptation mechanisms. In *Drosophila* and many other organisms, hypoxia (low oxygen) restricts systemic growth and reduces body size[3,18–22], and in humans the limited oxygen associated with high-altitude living has been linked to slow growth and developmental delay[23,24]. These effects occur at oxygen concentrations above those that compromise basic metabolism[25,26], indicating that they do not reflect limited aerobic respiration but rather an active adaptation under genetic control. The conserved transcription factor hypoxia-inducible factor 1 alpha (HIF-1a) is the key regulator required for these adaptive responses. At a rate proportional to oxygen levels, it is marked for degradation by HIF-1a prolyl hydroxylase (Hph)[27]. Thus, under hypoxic conditions, HIF-1a is able to perdure and (with its constitutively present beta subunit) induce target-gene expression. Although it is well established that nutrients mainly affect growth through insulin, the mechanism by which animals adaptively limit growth under oxygen limitation remains to be determined.

We describe here an RNA interference (RNAi)-based screen for body-size defects, covering genes encoding potential secreted factors and their receptors, in which we identify *breathless* (*btl*)[28], encoding a fibroblast growth factor receptor (FGFR) homolog, as one of the strongest hits. This receptor is expressed by cells of the tracheae, the gas-delivery tubules of the insect respiratory system. Its FGF-like ligand Branchless (Bnl) is produced by target tissues in a stereotyped pattern and in response to tissue hypoxia during development[29,30] to guide the tracheal terminal branches toward target tissues. When this system is perturbed, internal tissues experience hypoxia. We find that inducing tissue hypoxia by manipulating Btl-mediated tracheal outgrowth or external $O_2$ levels, or genetically inducing ectopic hypoxia responses via *Hph* or *HIF-1a* perturbation, alters the expression of *Dilp2*, -3, and -5 in the IPCs and blocks DILP release, leading to reduced systemic insulin signaling. We further report that the primary sensor of internal oxygen availability is the fat body. Both hypoxia and amino acid (AA) insufficiency block Hph activity in this tissue, and this effect propagates through two divergent pathways downstream of Hph: HIF-1a-independent inhibition of the Tor pathway alters adipose-tissue physiology, whereas a Tor-independent, HIF-1a-dependent pathway leads to the release of one or more humoral factor(s) that strongly inhibit DILP release and thereby adapt organismal growth to oxygen availability. Thus, Hph/HIF-1a and Hph/Tor pathways in the fat body function as central integrators of both oxygen and AA levels to adapt growth and metabolism to environmental conditions. Understanding the changes brought about by hypoxia in this model system may allow greater understanding of human disease associated with tissue hypoxia such as obesity and diabetes.

## Results

**In-vivo RNAi screen for signals affecting body size.** We undertook a genetic RNAi[31,32] screen targeting 1845 genes encoding the *Drosophila* secretome and receptome (Fig. 1a, Supplementary Data 1). Ubiquitous knockdown using the *daughterless-GAL4* driver (*da*>) identified 89 genes whose manipulation resulted in significant size phenotypes (Fig. 1b, c). Among these were several genes associated with body or tissue growth including *Insulin receptor* (*InR*)[33], *Anaplastic lymphoma kinase* (*Alk*)[34] and *jelly belly* (*jeb*)[35], *Epidermal growth-factor receptor* (*Egfr*)[36], and the prothoracicotropic hormone (PTTH) receptor *torso*[37], thus validating our approach. Genes important for ecdysone signaling (*knirps*, *Eip75B*, *Npc2g*, and *torso*[37–40]) were among the hits that increased pupal body size, consistent with the growth-limiting effects of ecdysone, whereas knockdown of the growth-promoting *InR* produced a strong decrease in size. We identified FGF-pathway signaling (Supplementary Fig. 1) among the strongest hits associated with reduced growth. The main hit, the *FGFR* ortholog *btl*, and its FGF-like ligand Bnl, are critical for proper tracheation of tissues to allow for gas exchange and oxygen delivery[28].

**FGF signaling affects systemic growth via tracheal effects.** Ubiquitous *btl* (*da* > *btl-RNAi*) and *bnl* (*da* > *bnl-RNAi*) knockdown strongly reduced growth and pupal size (Fig. 2a, b). Knockdown of the upstream transcription factor *trachealess* (*trh*)[41], also required for tracheal formation, reduced body size to a lesser degree, perhaps because of low RNAi efficiency or Trh-independent signaling later in development[42]. *btl* is mainly expressed in the tracheal system but it is also active in other cells[43]. To identify the tissue causing the size phenotype, we assayed effects of *btl* RNAi in the tracheae (*btl* > *btl-RNAi*), neurons (*elav*>), musculature (*Mef2*>), fat body (*ppl*>), and glia (*repo*>). Only tracheal knockdown of *btl* led to reduced body size (Fig. 2c). RNAi specificity was confirmed with two additional independent *btl-RNAi* lines (Supplementary Fig. 2a). Knockdown of *btl* in the tracheae delayed pupariation, thus prolonging the larval growth period, suggesting that small size resulted from reduced growth rate (Supplementary Fig. 2b). We therefore measured the growth rate during the third larval instar (L3) and found indeed that tracheal *btl* RNAi systemically slowed body growth (Fig. 2d). Thus, *btl* is required in the tracheal system for normal systemic growth, possibly via effects on oxygen delivery.

**The *btl-RNAi* size phenotype is mediated by insulin.** Among the primary regulators of systemic growth in *Drosophila* are the

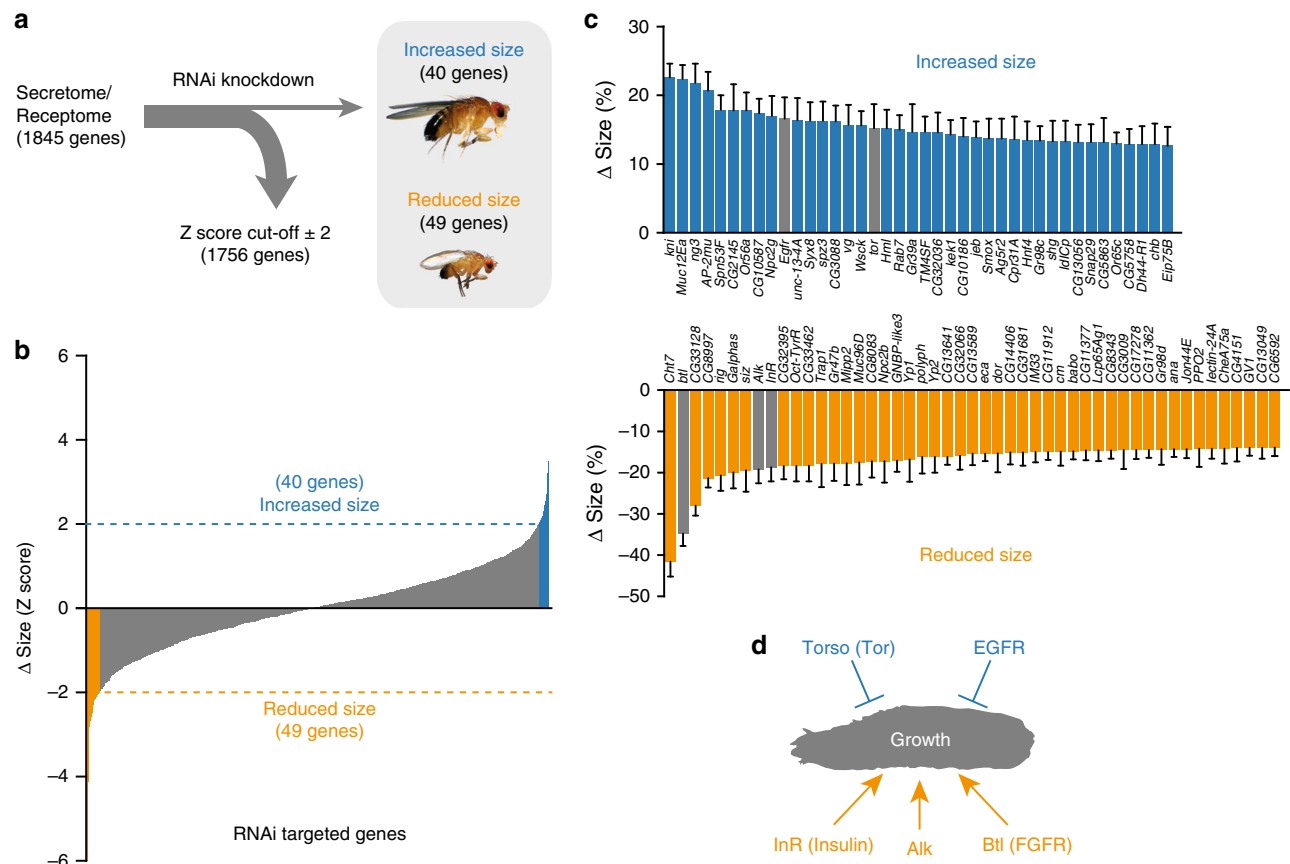

**Fig. 1** In-vivo RNAi screen for factors regulating body size. **a** RNAi screen of 1845 genes of the *Drosophila* secretome and receptome selected by gene ontology analysis. Upstream activating sequence (UAS)-inducible RNAi constructs against these genes were expressed ubiquitously using *da-GAL4*, and pupal sizes were determined (data are presented in Supplementary Data 1). **b**, **c** Distribution of the pupal sizes from the screen (**b**) revealed 89 gene hits that resulted in a significant increase (**c**: blue, top) or decrease (**c**: orange, bottom) in body size (a *Z*-score > ± 2 was used as a cutoff) by comparison with the mean of all lines. *n* = 9–55. **d** Genes and pathways identified in the screen having known size-regulatory function, as well as *breathless* (*btl*), are marked with arrows. Error bars indicate SEM. Underlying data are provided in Supplementary Data 1

DILPs[7]. To assess whether the body-size phenotype of *btl* knockdown is mediated by altered insulin signaling, we first examined the effects of directly reducing insulin signaling. As expected, globally reducing insulin signaling (*da* > *InR-RNAi*) or blocking Tor-dependent fat-body nutrient signaling through expression of *Tsc1* and *Tsc2* (*ppl* > *Tsc1/2*) strongly reduced pupal size and prolonged larval development (Fig. 3a, Supplementary Fig. 3a), phenocopying tracheal *btl* loss. To determine whether *btl* > *btl-RNAi* might cause insulin expression changes, we quantified *Dilp2*, *-3*, and *-5* transcripts in feeding L3s, ~ 24 h before pupariation. Indeed, very strong reduction (>90%) of *Dilp3* expression was observed in RNAi animals, accompanied by a smaller decrease in *Dilp5* transcripts (Fig. 3b).

Insulin activity is also regulated at the level of DILP release[44], and we therefore investigated whether tracheal *btl* loss might induce IPC retention of DILP2, -3, and -5. Knockdown induced an accumulation of these peptides in the IPCs (Fig. 3c), despite the reduced expression of *Dilp3* and *Dilp5*. This suggests that *btl* > *btl-RNAi* induces a strong decrease in DILP secretion, which should reduce peripheral activity downstream of InR. Insulin signaling leads to the phosphorylation of the kinase Akt and to suppression of *InR* expression as part of a sensitivity-regulating feedback circuit[45]. We indeed observed *InR* upregulation and phosphorylated Akt (pAkt) reduction in *btl* > *btl-RNAi* animals (Fig. 3d–f), indicating decreased insulin signaling, presumably arising from blocked IPC DILP release. To determine whether the observed reduction in insulin signaling was the direct cause of the

observed size phenotype, we asked whether rescuing circulating insulin levels by overexpressing *Dilp2* would also rescue growth. Ubiquitous *btl* knockdown driven by *armadillo-GAL4* (*arm*>) led to reduced body size that was rescued by *Dilp2* overexpression (Fig. 3g), supporting that the size phenotype is caused by a systemic reduction in insulin signaling downstream of Btl.

Alternatively, the observed reduced body size might involve increased ecdysone signaling, which antagonizes insulin signaling and suppresses larval body growth[46]. However, *btl* knockdown led to reduced expression of the ecdysone-induced gene *E75B*, which indicates reduced ecdysone-induced growth restriction (Supplementary Fig. 3b). Together, these results suggest that the reduced body size observed with *btl* knockdown is mediated by effects in the tracheal system that lead to altered expression and reduced secretion of DILP2, -3, and -5 from the IPCs, resulting in reduced insulin-driven systemic growth.

**btl RNAi leads to hypoxia via reduced tracheation.** Hypoxic tissues secrete the FGF-like ligand Bnl, inducing tracheal growth toward these tissues via its receptor Btl. To assess whether *btl* knockdown reduced tracheation of internal tissues, we observed tracheal branches associated with the fat body in larval abdominal segments 2 and 3 (Fig. 4a), quantifying the number of branches and total tracheal length[47]. Control tracheae exhibited extensive terminal branching, which was significantly reduced in *btl* > *btl-RNAi* animals, along with the total length of tracheal processes

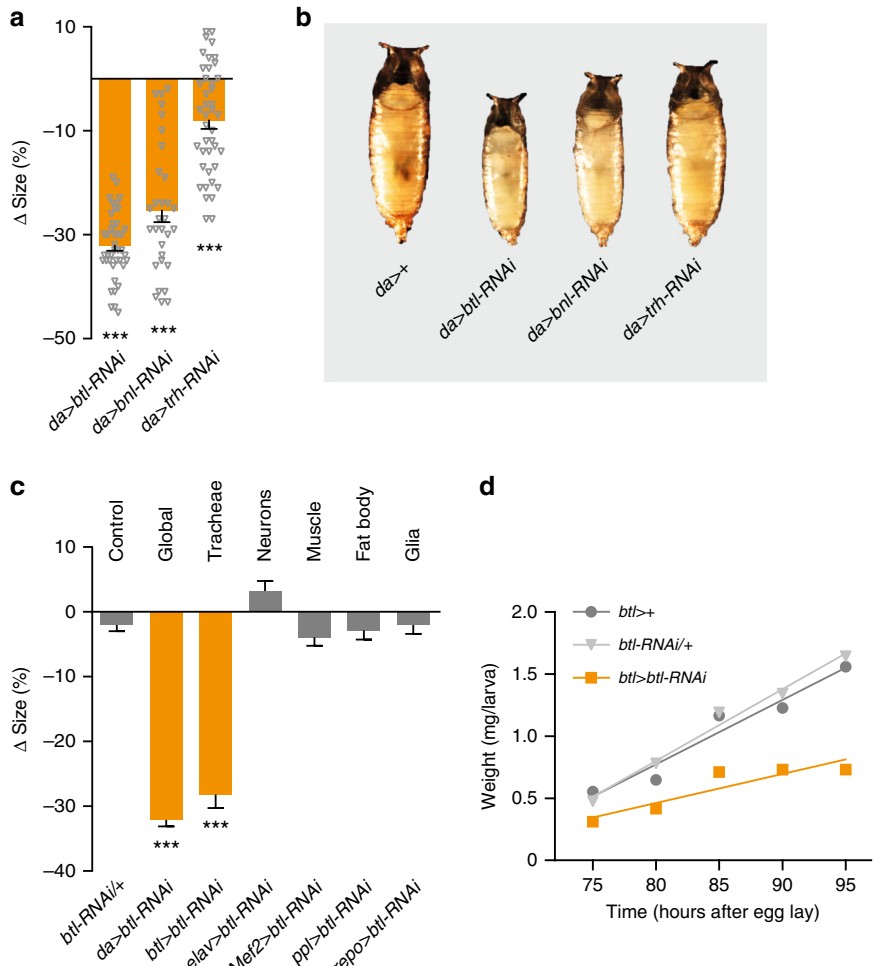

**Fig. 2** Tracheal loss of *breathless* reduces systemic growth and body size. **a**, **b** Quantification of pupal sizes and representative images (**b**) of animals with ubiquitous knockdown of *breathless* (*btl*)-pathway components compared with the controls (*da > +*; *da >* crossed to wild type, *w[1118]*). Values are percent change in pupal size vs. control. **a**: $n = 31$–64. **c** Knockdown of *btl* in whole animals (*da > btl-RNAi*) or tracheae alone (*btl>*), but not in neurons (*elav>*), muscle (*Mef2>*), fat-body tissue (*ppl>*), or glia (*repo>*) leads to small animals compared with *da > +* controls. Shown also is the RNAi control (*btl-RNAi/+*; *btl-RNAi* crossed to *w[1118]*). Values are percent change in pupal size vs. the *btl > +* controls (*btl >* crossed to *w[1118]*). $n = 33$–86. **d** Trachea-specific knockdown of *btl* reduces larval growth rates compared with driver (*btl > +*) and RNAi controls (*btl-RNAi/+*). Statistics: one-way ANOVA with Dunnett's multiple-comparisons test. ***$P < 0.001$, compared with the control. Error bars indicate SEM. Underlying data are provided in the Source Data file

(Fig. 4b, c), implying a reduced oxygen supply. In *btl > btl-RNAi* animals, we observed strong upregulation of *bnl* (Fig. 4d), which is induced in response to hypoxia[29], indicating that internal tissues were indeed experiencing hypoxia. We rationalized therefore that these animals would be sensitized to low-oxygen conditions. To test this, we reared control and *btl > btl-RNAi* animals under hypoxic conditions (5% $O_2$ vs. the normal 21% atmospheric level) and measured their survival to the pupal stage. Most control animals survived this environment, reflecting their ability to adapt to reduced oxygen levels, whereas none of the *btl-RNAi* animals survived to pupariation (Fig. 4e), presumably dying from hypoxia-induced damage. Taken together, our data suggest that the insulinergic changes described above operate downstream of tissue hypoxia to adapt growth to oxygen availability.

To investigate whether insulinergic changes might regulate the tracheae themselves, we assessed the effects of altering insulin signaling in these tissues. Although overexpression of *Dilp2* rescued growth of *arm > btl-RNAi* animals (Fig. 3g), it did not restore the growth of their tracheae (Supplementary Fig. 3c). Furthermore, tracheal knockdown of *InR* or *Akt* had no effect on systemic growth or pupal size, whereas loss of Tor signaling in the tracheae led to a strong reduction in size (Supplementary Fig. 3d).

Thus, our findings place insulin downstream of the growth of the tracheal system and oxygen levels, mediating their effects on systemic growth.

**Hypoxia pathway activation mimics loss of *btl*.** Thus, ubiquitous or trachea-specific loss of *btl* induces internal tissue hypoxia, which via reduced insulin signaling leads to reduced organismal growth. To confirm that hypoxia per se is the cause of this reduction and to assess whether this inhibition is part of an adaptive response rather than a pathological result of low oxygen levels, we investigated the effects of external hypoxia on larval growth. Wild-type (*w[1118]*) animals reared under 5% $O_2$ displayed reduced pupal size, despite a delay in pupariation that extended the larval growth period, phenocopying *btl > btl-RNAi* and indicating strong growth reduction (Fig. 5a, b, Supplementary Fig. 4a).

The signaling pathway that transduces oxygen levels to induce cellular adaptation is conserved between *Drosophila* and mammals. The main effector protein of this pathway is the transcription factor HIF-1a, encoded in *Drosophila* by the gene *similar* (*sima*)[48]. Under normoxic conditions, this protein is

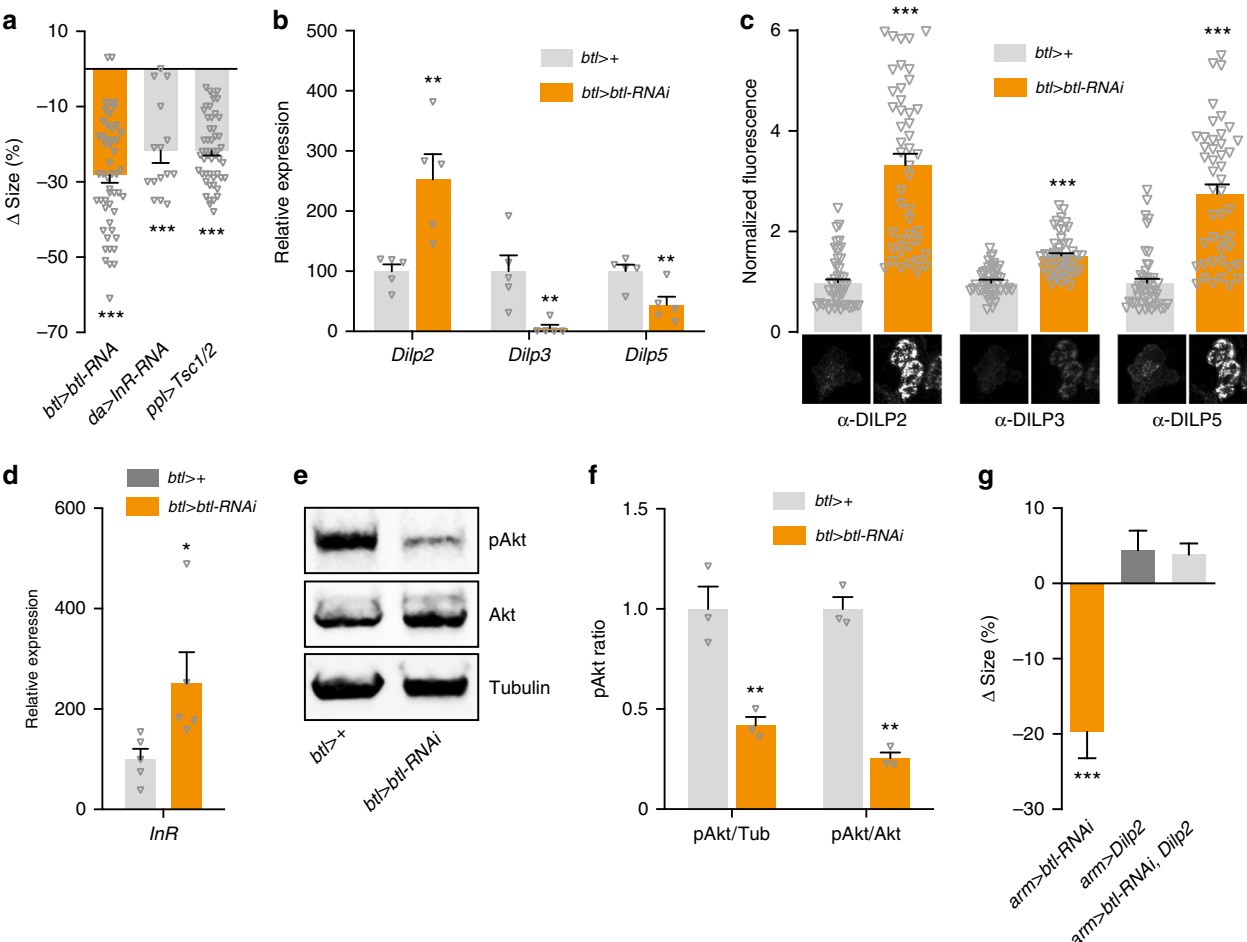

**Fig. 3** Breathless modulates systemic growth by altering DILP signaling. **a** Trachea-specific knockdown of *breathless* (*btl*) mimics loss of insulin signaling induced by ubiquitous *InR* knockdown (*da > InR-RNAi*) and fat-body inhibition of Tor (*ppl > Tsc1/2*), which remotely inhibits release of insulin-like peptides (DILPs) from the insulin-producing cells (IPCs). Values are percent change in pupal size vs. the *btl >* + controls (*btl>* crossed to *w1118*). $n = 15$–79. **b**, **c** Levels of whole-animal *Dilp* transcripts and DILP peptides in the IPCs in *btl > btl-RNAi* animals with knockdown of *btl* in the trachea compared with *btl >* + controls. Representative images of DILP2, -3, and -5 immunostainings are shown below. **b**: $n = 5$. **c**: $n = 44$–45. **d** Increased expression of *InR* in *btl > btl-RNAi* animals compared with *btl >* + controls suggests reduced systemic insulin signaling. $n = 5$. **e**, **f** Immunoblotting showing whole-animal levels of phosphorylated Akt (pAkt) (**e**) in *btl > btl-RNAi* animals compared with *btl >* + controls, quantified in **f** from three independent replicates normalized to Tubulin (Tub) or Akt levels. **f**: $n = 3$. **g** Ectopic expression of *Dilp2* rescues the *btl*-knockdown size phenotype. Ubiquitous weak knockdown of *btl* using *armadillo-GAL4* (*arm > btl-RNAi*) leads to small pupae (left). Co-expressing *Dilp2* rescues this size phenotype (right), while having no significant effect on control animals (middle). Values are percent change in pupal size versus *arm >* + controls. $n = 14$–113. Statistics: one-way ANOVA with Dunnett's test for multiple comparisons and Student's *t*-test for pairwise comparisons. *$P < 0.05$, **$P < 0.01$, ***$P < 0.001$, compared with the control. Error bars indicate SEM. Underlying data are provided in the Source Data file

marked for proteasomal degradation through the oxygen-dependent activity of Hph, also called Fatiga (Fga) in *Drosophila*[49,50]. To differentiate between pathological damage vs. a genetic adaptive response, we activated the hypoxia-adaptation program by mutation of *Hph*, thus de-repressing Sima under normoxic conditions. Similar to hypoxia or *btl* knockdown, loss of *Hph* function resulted in growth inhibition and developmental delay despite the animals' normoxic environment (Fig. 5a, b, Supplementary Fig. 4a), confirming the genetic control of the growth phenotype.

To evaluate whether the insulin-related changes seen in *btl*-knockdown animals are part of a genetic hypoxia-adaptation program, we measured *Dilp* expression and DILP retention within the IPCs in hypoxic wild-type animals and in normoxic *Hph* mutants. *Dilp3* and *Dilp5* transcription was strongly reduced in both manipulations (Fig. 5c). Furthermore, both conditions led to increased IPC DILP2 and DILP5 retention (Fig. 5d). The observed DILP3 level may represent retention of this protein as

well, as *Dilp3* expression is greatly reduced by up to 90%. To confirm IPC DILP retention, we expressed epitope-tagged DILP2[51] in these cells using *Dilp2-GAL4* (*Dilp2>*) and measured levels of circulating tagged insulin by enzyme-linked immuno-sorbent assay (ELISA). Starvation conditions, known to reduce hemolymph DILP2 levels[52], greatly reduced tagged DILP2 titer (Fig. 5e). However, feeding animals moved to hypoxia showed an even stronger decrease in insulin release. Thus, hypoxia indeed blocks DILP release from the IPCs. Consistent with decreased systemic insulin signaling, both the low-O$_2$ environment and *Hph* loss led to reduced pAkt levels (Fig. 5f, Supplementary Fig. 4b). We further investigated this using an in-vivo insulin-pathway reporter that reflects intracellular insulin-pathway activity based on plasma-membrane localization of green fluorescent protein (GFP)[53]. Under normoxia GFP was localized to the membrane of fat-body cells, indicating active insulin signaling, whereas under hypoxia, the reporter remained cytoplasmic, confirming reduced signaling (Fig. 5g).

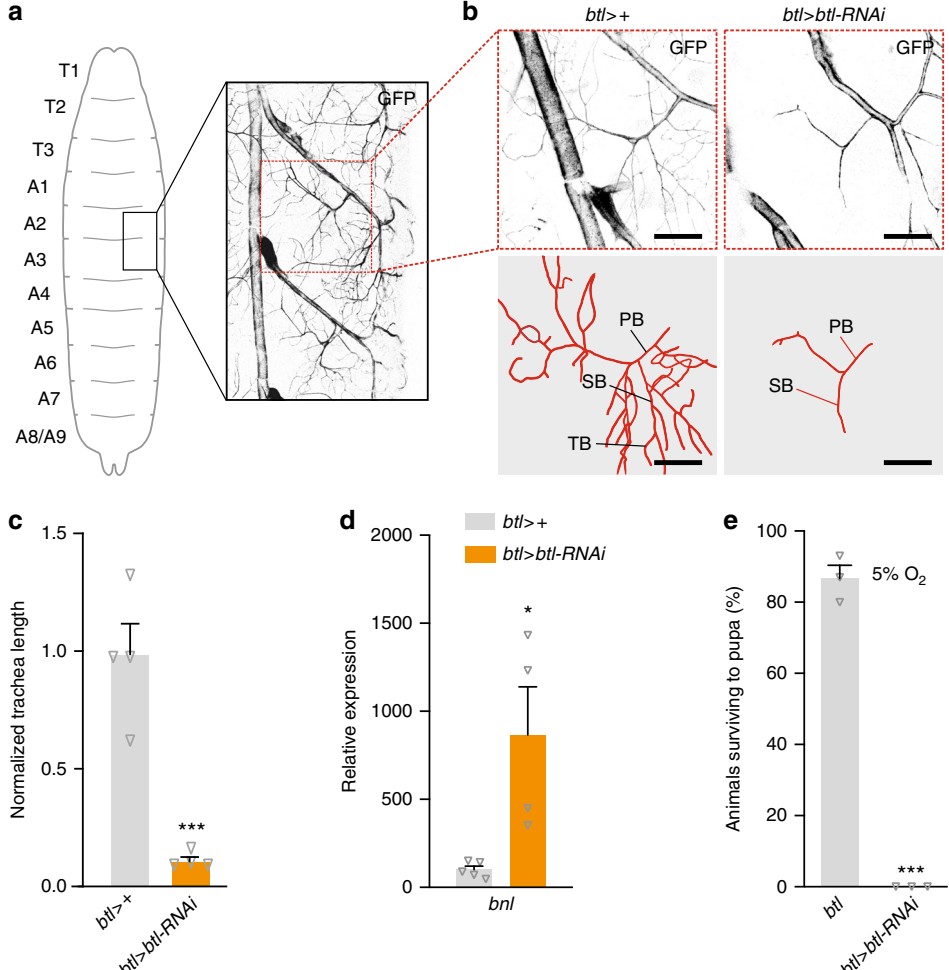

**Fig. 4** Airway branching is reduced in *breathless*-knockdown animals. **a** Tracheae ramifying to the fat body were measured in larval segments A2/A3, illustrated in overview here. **b, c** Knockdown of *breathless* (*btl*) in tracheae leads to a reduction in branching of the terminal tracheal cells of the fat body branch compared with the *btl* > + control (*btl* > crossed to $w^{1118}$). Representative images (**b**) and quantification (**c**) of the fat-body branch of larval tracheal terminal cells are shown. Scale bar, 50 μm. PB, primary branch; SB, secondary branch; TB, tertiary branch. **c**: *n* = 4. **d** Whole-body transcript levels of *branchless* (*bnl*), encoding the hypoxia-inducible FGF ligand of Btl, is strongly upregulated in larvae with trachea-specific *btl* knockdown, suggesting that tissues are experiencing hypoxia in these animals. *n* = 4–5. **e** Loss of *btl* renders animals unable to survive in a low-oxygen environment (5% O₂ levels), whereas most *btl* > + controls are able to develop to pupariation. *n* = 3. Statistics: Student's *t*-test for pairwise comparisons. *$P < 0.05$, ***$P < 0.001$, compared with the control. Error bars indicate SEM. Underlying data are provided in the Source Data file

Similar to animals with *btl* knockdown, hypoxic wild-types and normoxic *Hph* mutants exhibited reduced ecdysone signaling (measured as expression of *E75B*) and upregulation of *bnl* (Supplementary Fig. 4c, d). To rule out ecdysone deficiency as an underlying cause of the hypoxic growth effect, we supplemented wild-type animals with ecdysone by feeding and found that this treatment even further decreased pupal sizes under hypoxic conditions (Supplementary Fig. 4e), demonstrating that reduced ecdysone does not underlie hypoxia-induced growth restriction. Taken together, these findings are consistent with an underlying phenomenon common to *btl* knockdown, hypoxic wild-types, and normoxic *Hph* mutants—namely, the activation of the genetic hypoxia-adaptation program that limits growth by reducing insulin signaling.

**Loss of *btl* or low O₂ cause adipose-tissue hypoxia.** The fat body is reported to secrete factors that modulate IPC insulin expression and release[9–11,14–17]. To investigate whether this tissue is also involved in hypoxia adaptation, we first measured fat-body *bnl* transcription under tracheal *btl* knockdown and observed strong

elevation, indicating that this tissue experiences hypoxia with tracheal insufficiency (Fig. 6a). To confirm this hypoxic state, we made use of a genetic hypoxia indicator based on GFP fused with the oxygen-dependent degradation domain (ODD) of HIF-1a[54]. Under normoxic conditions, this GFP-based reporter is degraded through Hph activity, whereas red fluorescent protein (RFP) expressed from the same regulatory sequences acts as a ratiometric control. In the fat body, the GFP::ODD reporter was rapidly degraded under normoxia, leading to a low GFP:RFP ratio (Fig. 6b, Supplementary Fig. 5a). In contrast, an increased ratio was observed in animals reared under 5% O₂, indicating reduced oxygen-dependent degradation of the ODD-linked GFP. Together, these data show that the fat body experiences physiologically significant oxygen stress under our manipulations.

**An adipose-derived hypoxia signal represses insulin release.** We next asked whether the brain and the IPCs alone could sense hypoxia and induce changes in insulin physiology, or whether the fat body might be necessary for organismal growth adaptation to hypoxia. To investigate this, we cultured larval brains alone or with

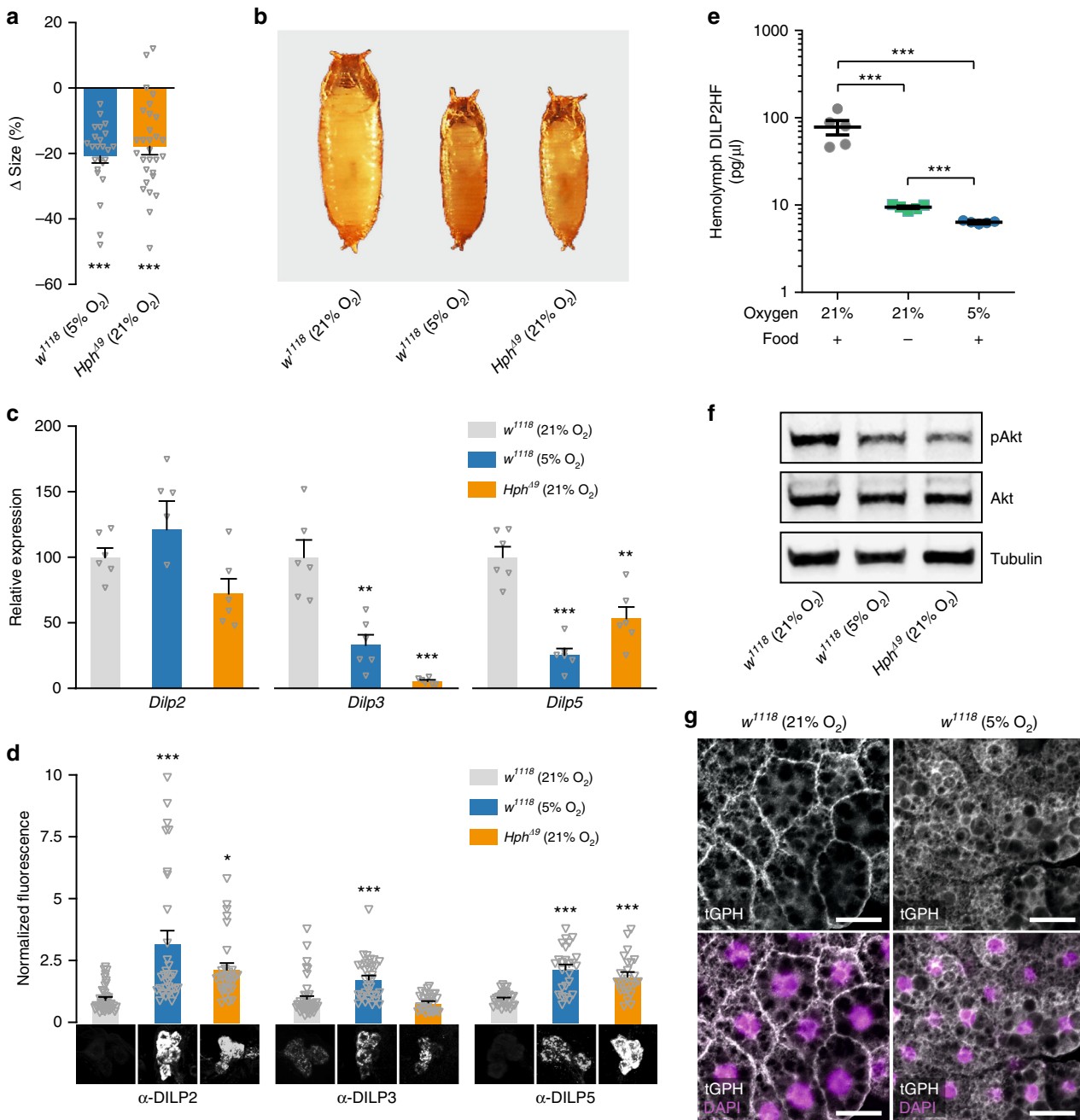

**Fig. 5** Hypoxia and induction of hypoxia adaptation block growth and DILP secretion. **a**, **b** Quantification of pupal size (**a**) and representative images (**b**) of animals exposed to hypoxia (5% $O_2$) or mutation of HIF-1a prolyl hydroxylase (*Hph*) compared with controls (*w1118* in normoxia). Values are percent change in pupal size vs. *w1118*-in-normoxia controls. **a**: $n = 24$–34. **c**, **d** Transcript levels of *Dilp* genes in whole animals (**c**) and DILP peptide levels (**d**) in the insulin-producing cells (IPCs) of hypoxic wild-types and normoxic *Hph* mutants compared with normoxic wild types. Representative images of IPC DILP2, -3, and -5 immunostaining are shown below. **c**: $n = 6$. **d**: $n = 29$–42. **e** Hemolymph HA::DILP2::FLAG (DILP2HF) levels in normoxic (21% $O_2$, on normal food), hypoxic (5% $O_2$; normal food), and starved (21% $O_2$; 1% agar) animals determined by ELISA. $n = 5$. **f** Immunoblotting shows that levels of phosphorylated Akt (pAkt) is reduced under hypoxia and in *Hph* mutants compared with *btl* > + controls when normalized to alpha-Tubulin (Tub) or total Akt levels. **g** tGPH reporter of insulin signaling confirms that hypoxia reduces systemic insulin signaling in peripheral tissues, represented here by the fat body. Under conditions of normal insulin activity (e.g., fed normoxia, left), the tGPH sensor is localized to the plasma membrane, whereas under conditions of low signaling (e.g., fed hypoxia, right) the sensor becomes primarily cytoplasmic. Scale bar, 20 μm. Statistics: one-way ANOVA with Dunnett's multiple-comparisons test. *$P < 0.05$, **$P < 0.01$, ***$P < 0.001$, compared with the control. Error bars indicate SEM. Underlying data are provided in the Source Data file

fat bodies for 16 h ex vivo under normoxia or hypoxia and assessed the insulin-related effects of these manipulations (Fig. 6c). Whereas hypoxic larvae exhibit reduced *Dilp3* and *-5* expression (Fig. 5c), we observed no decrease in insulin-gene expression in brains cultured alone under hypoxia vs. normoxia (Fig. 6d). Brains co-cultured with fat bodies, however, did exhibit reduced *Dilp* expression under hypoxia compared with normoxic cultures. These observations demonstrate that the brain alone is insufficient for hypoxia adaptation, and that a humoral signal(s) from the fat body regulates *Dilp3* and *-5* expression as part of this response.

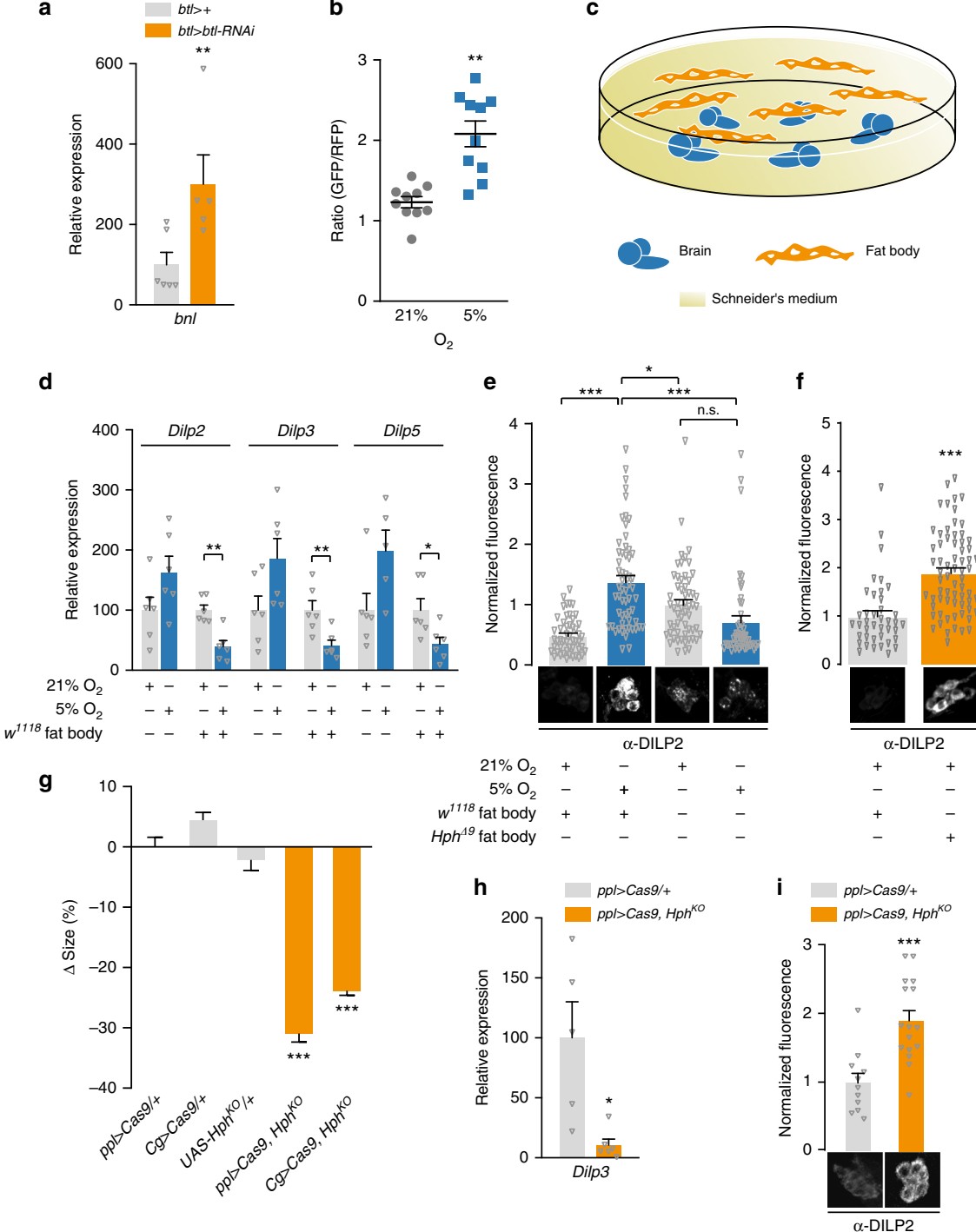

**Fig. 6** A fat-body hypoxia sensor represses insulin secretion from the brain. **a** Transcriptional upregulation of *branchless* (*bnl*) in the fat body of animals with trachea-specific *btl* knockdown indicates hypoxia. $n = 5$–6. **b** A transgenic reporter of hypoxia indicates that the fat body experiences low oxygen levels that inhibit Hph when animals are incubated under 5% $O_2$ compared with 21% $O_2$ conditions. Ratio of integrated GFP (carrying the oxygen-dependent degradation domain of HIF-1a) and RFP (without this domain) signals in fat body. $n = 10$. **c** Illustration of ex-vivo organ culture technique. **d** Transcript levels of *Dilp* genes in brains alone or brains and fat bodies cultured ex-vivo under hypoxia, compared with normoxic cultures. *Dilp* transcription was unchanged or increased (nonsignificant) when brains alone were exposed to hypoxia, whereas transcription of all three genes was reduced in brains co-incubated with fat-body tissue, closely mimicking the changes seen in whole animals. $n = 5$–6. **e** Ex-vivo co-culture of wild-type brains with wild-type fat bodies causes DILP2 retention under hypoxia. Brains incubated alone without fat body are unable to induce DILP2 retention under hypoxia. Thus, the presence of the hypoxic fat body actively represses DILP secretion. $n = 50$–58. **f** Co-culture of wild-type brains with *Hph*-mutant fat bodies induces DILP2 retention under normoxia (right). Representative images of DILP2 immunostainings in the IPCs are shown below. $n = 40$–67. **g-i** CRISPR/Cas9-induced fat-body-specific *Hph* knockout (*Hph*$^{KO}$) reduced pupal size (**g**) and transcript levels of *Dilp3* (**h**), and caused DILP2 retention in the IPCs (**i**). Representative images of DILP2 immunostainings in the IPCs are shown below. The *ppl>* and *Cg>* fat-body GAL4 drivers were used to express UAS-driven *Cas9* and gRNAs designed to delete part of the *Hph* locus specifically in the fat body. Controls: *ppl > Cas9/+*, *Cg > Cas9/+*, and *UAS-Hph*$^{KO}$/+ (*Hph*$^{KO}$/+). **g**: $n = 23$-44. **h**: $n = 5$-6. **i**: $n = 11$-16. Statistics: Student's *t*-test for pairwise comparisons and one-way ANOVA with Dunnett's for multiple-comparisons test. *$P < 0.05$, **$P < 0.01$, ***$P < 0.001$, compared with the control. Error bars indicate SEM. Underlying data are provided in the Source Data file

To confirm the requirement of fat-body genetic hypoxia responses for DILP2 retention, we cultured wild-type brains either alone or with wild-type or *Hph*-mutant fat bodies, under normoxia and hypoxia. Co-culturing wild-type brains and fat bodies under normoxia induced a drop in DILP2 retention compared with brains alone, consistent with previous studies showing that the fat body releases humoral signals that promote insulin secretion in well-fed animals reared under normal oxygen conditions[9,12,16,55]. Brains cultured alone did not exhibit DILP2 changes between hypoxia and normoxia; however, when brains were cultured with wild-type fat-body tissue, hypoxia strongly induced DILP2 retention (Fig. 6e). These changes indicate that hypoxia-induced inhibitory fat-body factor(s) are required for insulin retention. To assess the genetic control of this signal, we co-cultured wild-type brains with *Hph*-mutant fat bodies under normoxia (Fig. 6f) and found again that DILP2 was retained in the IPCs.

Next, as this fat-body oxygen sensor regulates insulin physiology, we wished to examine its necessity in the regulation of systemic growth. To ensure efficient gene-activity disruption, we made UAS-inducible CRISPR/Cas9 constructs to delete *Hph* specifically in the fat tissue. Under normoxia, pupal size was strongly reduced by fat-body-specific disruption of *Hph*, driven by either of two independent fat-body-*GAL4* lines (Fig. 6g), similar to the size reduction observed with hypoxia or traditional *Hph* mutation. Furthermore, we found that deletion of *Hph* in the fat body led to reduced *Dilp3* transcription and to DILP2 retention in the brain (Fig. 6h, i) under normoxia. Fat-body-specific *Hph* knockout therefore recapitulates major components of the organismal hypoxic response. Taken together, these results indicate that the activity of the Hph-regulated hypoxia-adaptation program within the fat body is the determinant, via secreted inhibitory factor(s), of IPC insulin retention and thereby of systemic growth rate.

**Hph is required for Tor activity, independently of HIF-1a**. Beyond its role in coupling oxygen to systemic growth demonstrated above, the fat body also links intake of dietary AAs to systemic growth through a Tor-dependent nutrient-sensing mechanism that regulates release of insulin-governing factors. We therefore next explored the possible relationships between oxygen and AA sensing in growth regulation. We reared wild-type animals and *Hph* and *sima* mutants[56] on foods containing 0.1×, 0.3×, or 1× (standard food) concentrations of yeast (the customary dietary protein source) under normoxia or 5% $O_2$ and assessed the animals' growth responsiveness to dietary protein. Whereas wild-type animals displayed a robust scaling of body size to dietary protein levels under normoxia, hypoxia attenuated or blocked this response (Fig. 7a). Similarly, inducing genetic hypoxic responses by *Hph* mutation rendered animals growth-unresponsive to protein even under normoxia, indicating that normal growth responses to dietary AAs require Hph signaling. To address whether this effect requires HIF-1a, we analyzed the growth response of *HIF-1a/sima* mutants to dietary protein. Under normoxic conditions, *sima*-mutant animals responded to dietary AAs to the same degree as wild types. Thus, unlike Hph, Sima is not required for AA-dependent growth regulation. However, in contrast to wild-type animals, *sima* mutants still responded to dietary protein even under hypoxic conditions, and their hypoxia-induced size reduction was significantly rescued (Fig. 7a), indicating that Sima is required for hypoxia-induced growth restriction.

As a number of known growth-regulatory factors released by the fat body are regulated by Tor activity in response to AA availability[10,11,16,17], we investigated whether Tor signaling might

be altered by hypoxia, and, if so, how AA levels and Hph and Sima/HIF-1a activity affected this alteration. We reared wild-type animals on standard 1× food in normoxia and hypoxia, and measured levels of phosphorylated ribosomal protein S6 (pS6), a downstream effector and proxy for Tor signaling[57], in the fat bodies of feeding late-third-instar larvae. Hypoxic rearing very strongly reduced pS6 levels (Supplementary Fig. 5b), indicating that constitutive hypoxia induces a large reduction in Tor activity. We next measured levels of pS6 in wild-type animals and *Hph* and *sima* mutants, under normoxia or 16 h hypoxic conditions, with and without 16 h AA starvation. Wild-type animals kept on standard 1× food under normoxia exhibited high levels of pS6, indicating strong Tor activity (Fig. 7b), and 16 h hypoxia treatment led to a decrease in pS6 staining, as did AA starvation, which is known to inhibit Tor. Combining these treatments, hypoxia and AA starvation, did not further reduce fat body pS6 levels beyond that of starvation alone, suggesting either that they act through the same pathway or that Tor is already maximally inhibited by protein starvation.

*Hph*-mutant animals exhibited low fat-body pS6 staining that was not affected by oxygen or AA levels. This is consistent with the hypoxic response as well as the lack of growth response of *Hph* mutants to dietary AAs, indicating that Hph activity is required for Tor-pathway function. A known feedback loop connects HIF-1a/Sima to Tor inhibition[58,59], so we next investigated whether Sima is required for hypoxia-induced Tor inhibition. In contrast to loss of *Hph*, animals lacking *sima* still exhibited high levels of Tor activity under standard food and normoxic conditions, indicating that Sima is not required for Tor activity in these conditions. Furthermore, these animals also retained Tor inhibition under restrictive conditions of oxygen or AA deficiency (or both), indicating that Sima is not required for Tor inhibition in these situations. Therefore, oxygen-regulated Hph activity, but not Sima activity, is required for Tor activity and response to AAs.

To assess the physiological significance of Tor activity in the fat, we measured the size of lipid droplets (LDs), a parameter thought to reflect Tor activity[55], under the same environmental conditions as above. LDs of wild types were relatively small and their sizes increased under hypoxia or AA starvation (Fig. 7c), consistent with the reduced Tor activity observed under these conditions (Fig. 7a). On the other hand, loss of *Hph* led to greatly enlarged LDs in the fat body and their size was not further altered by hypoxia or AA starvation, consistent with the data above that suggest that Hph is required for transducing these signals through a Tor-dependent mechanism. Both hypoxia and AA starvation still led to increased LD size in *sima* mutants, indicating that Sima activity is not required for these effects, consistent with the finding that these animals also inhibit Tor appropriately in response to AA starvation and hypoxia. Thus, the Hph-dependent and HIF-1a/Sima-independent Tor-activity differences observed under varied environmental and genetic perturbations lead to physiological changes in fat distribution within the fat body.

As our data indicate that Tor-dependent AA responses require Hph activity, we next tested directly whether Hph might be involved in sensing AA levels, using the GFP::ODD Hph-activity reporter construct, and observed that 4 h protein starvation led to a significant increase in fat-body GFP::ODD perdurance (Fig. 7d). As Hph activity targets ODD-containing proteins for destruction, this result suggests that Hph activity is reduced by insufficiency of AAs as well as of oxygen. Thus, both AAs and oxygen may induce growth through modulating Hph activity, which is required for growth.

As our data suggest that both AAs and oxygen affect Tor through Hph, and Tor in the fat body is known to regulate the release of factors that control DILP secretion, we wished to

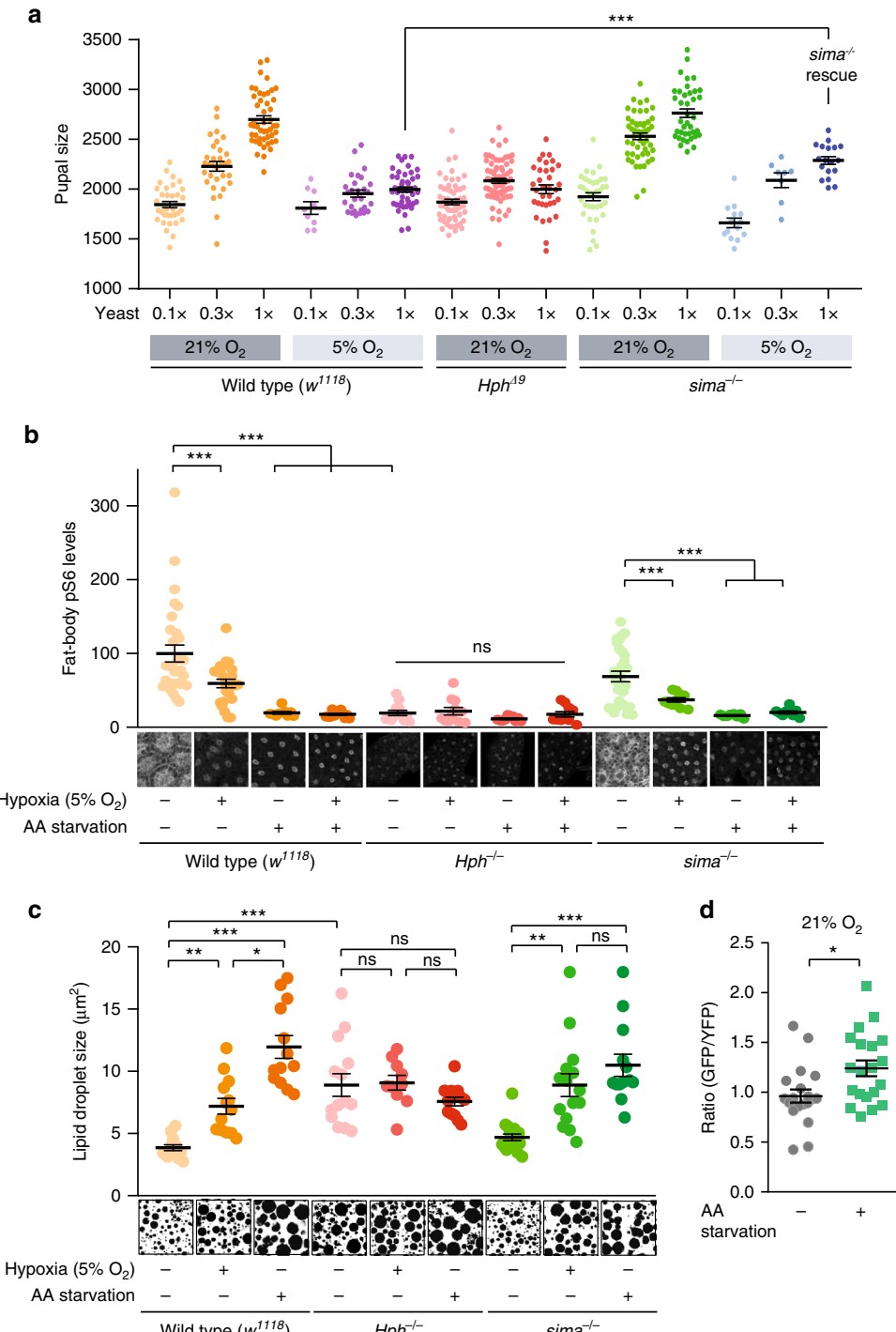

**Fig. 7** Hph-dependent amino-acid and oxygen sensing via Tor is HIF-1a/Sima-independent. **a** Growth response to dietary amino-acid levels (0.1×, 0.3×, and standard 1× dietary yeast concentrations) of wild types ($w^{1118}$) and Hph and sima mutants under normoxia (21% $O_2$) and hypoxia (5% $O_2$). Note that sima mutants are significantly larger than wild types on 1× yeast food in hypoxia, whereas their sizes are not significantly different in normoxia ($p = 0.26$), further indicating that Sima is required for hypoxia-induced growth suppression. $n = 8–71$. **b** Fat-body staining against phosphorylated ribosomal protein S6 (anti-pS6), a downstream indicator of Tor activity, indicates that the Tor pathway is suppressed by 16 h amino acid starvation (1× sugar-only medium), hypoxia (5% $O_2$), or Hph mutation. sima mutants exhibit Tor-pathway activation under normoxia, and thus Sima is not required for Tor activity. Tor activity decreases under hypoxia and with amino-acid deprivation, indicating that Sima activity is also not required for Tor suppression. Combined treatments of amino-acid starvation and hypoxia do not further suppress Tor. Representative images of pS6 stain are shown. $n = 8–30$. **c** Sizes of fat-body lipid droplets examined by CARS microscopy in wild types ($w^{1118}$) and Hph and sima mutants with 16 h hypoxia (5% $O_2$) or amino acid starvation (1× sugar-only medium). Lipid-droplet size increases under hypoxia and protein starvation in wild types and sima mutants, but not Hph mutants. Representative CARS images are shown. $n = 10–18$. **d** Hph activity reflects amino acid availability. Fat-body imaging of the GFP::ODD Hph-activity reporter indicates that GFP::ODD degradation is reduced under 4 h amino acid starvation (1x sugar food), suggesting that Hph activity is blocked by lack of amino acids. $n = 20$. Statistics: Student's *t*-test for pairwise comparisons and one-way ANOVA with Dunnett's for multiple-comparisons test. *$P < 0.05$, **$P < 0.01$, ***$P < 0.001$, compared with the control. Error bars indicate SEM. Underlying data are provided in the Source Data file

determine whether Tor activity per se might contribute to observed hypoxia-induced growth effects. We therefore stimulated the Tor pathway in the fat body by expressing the activator Rheb (*ppl > Rheb*). Although this manipulation dramatically increased Tor activity under hypoxia, it did not rescue the size phenotype (Supplementary Fig. 6a, b), indicating that the size-regulatory effects of hypoxia do not arise from alteration of Tor activity in the fat body. To further confirm the Tor independence of hypoxic growth reduction, we assessed the effects of individual known Tor-regulated growth modulators. Of the known fat-body-derived factors, Egr is the only one that is reported to be insulinostatic, although its specific effects do not exactly match those we observed under hypoxia[17]. We expressed RNAi against *egr* in the fat body or against the gene encoding its receptor, Grindelwald (Grnd), in the IPCs. Consistent with the idea of a Tor-independent mechanism, we found that neither *egr* nor *grnd* RNAi rescued the low-oxygen body-size reduction, suggesting that they do not play a role in hypoxia-induced growth restriction (Supplementary Fig. 6b). We also assessed other fat-body-derived factors and their receptors such as CCHa-2, Sun, GBPs 1 and 2, and Daw, although these are reported to be growth-promoting

factors, to rule out their involvement. None of these factors appears to affect hypoxia-induced growth restriction. Thus, the factor released by the fat body appears to be a previously uncharacterized Tor-independent signal that is released downstream of Hph and HIF-1a/Sima as an inhibitor of insulin secretion.

**Hypoxia reduces systemic growth via HIF-1a in the fat body.** As the data above suggested that HIF-1a/Sima is required in the fat body for hypoxia-adaptation effects on body size, we investigated the effects of fat-body-specific manipulations of HIF-1a/Sima function. First, we examined whether fat-body-specific RNAi against *sima* could block the DILP retention induced by hypoxia. We observed a small but significant reduction in IPC DILP2 levels and partial rescue of the hypoxia-induced growth inhibition with this manipulation (Supplementary Fig. 6c, d), which might indicate weak knockdown. Therefore, we repeated this experiment using tissue-specific CRISPR/Cas9-mediated disruption of *sima* and observed a much stronger reduction in IPC DILP2 levels under hypoxia (Fig. 8a). Furthermore, *Dilp3*

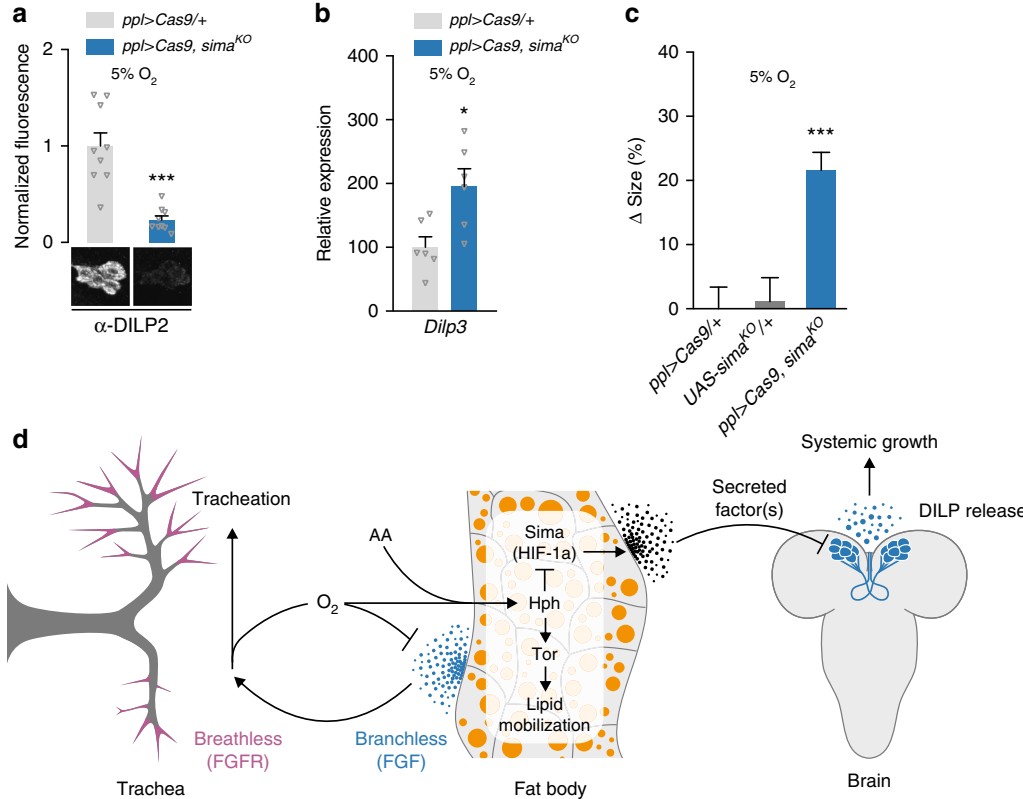

**Fig. 8** HIF-1a/Sima is required in the fat body for hypoxia-induced growth suppression. **a–c** Fat-body-specific loss of *sima* via tissue-specific CRISPR/Cas9-mediated gene disruption reduces hypoxia-induced IPC DILP2 retention (**a**) and suppression of *Dilp3* expression (**b**) and rescues hypoxia-induced growth restriction (**c**). Statistics: Student's *t*-test for pairwise comparisons and one-way ANOVA with Dunnett's for multiple-comparisons test. *$P < 0.05$, ***$P < 0.001$, compared with controls (*ppl > Cas9/+* and *UAS-sima^{KO}/+*). Error bars indicate SEM. Underlying data are provided in the Source Data file. **a**: $n = 9$. **b**: $n = 6$. **c**: $n = 18–33$. **d** Proposed model for integration of hypoxia and amino-acid sensing via the fat body and insulin regulation. Tracheation of the fat body is regulated by the FGF ligand Branchless secreted from this tissue in response to local tissue hypoxia, triggering tracheal outgrowth via the FGF receptor Breathless. When the tracheal system is unable to deliver sufficient oxygen because of environmental oxygen levels or body growth, Hph is unable to trigger the degradation of Sima/HIF-1a, which induces the expression or release of one or more humoral factors from the fat tissue that act on the insulin-producing cells (IPCs) to downregulate *Dilp* transcription and to block DILP release. Through a separate pathway, Hph inhibition also blocks Tor activity in the fat body, which alters lipid distribution. Hypoxia thereby reduces circulating insulin through activation of a central oxygen sensor in the fat tissue, which slows the growth of the whole organism. During these low-oxygen conditions, secretion of Branchless from the fat tissue promotes insulin-independent growth of the tracheal airways to increase oxygen supply. This system allows the organism to adapt its metabolism and growth to limited-oxygen conditions by reducing overall body growth through downregulation of insulin signaling, while at the same time promoting development of the tracheal system to maintain oxygen homeostasis

transcription was de-repressed (Fig. 8b) and the hypoxia-induced body-size reduction was significantly rescued by knockout of *sima* in the fat body (Fig. 8c). Taken together, our data show that fat-body HIF-1a/Sima activity is required for hypoxia-induced suppression of body growth via humoral signaling from the fat that inhibits insulin secretion.

## Discussion

In this report, we identify one tissue in particular, the fat body, which senses internal oxygen levels and regulates growth rate accordingly. Our data show that, as an adaptive response to oxygen limitation, the fat tissue releases into the circulation one or more factors that inhibit the secretion of insulin from the brain to reduce systemic growth (Fig. 8d). We demonstrate that the ability of oxygen to reduce systemic body growth through downregulation of insulin signaling requires Hph-dependent HIF-1a/Sima activity in the fat tissue. Furthermore, we show that hypoxia and AA deprivation both reduce Hph activity in the fat tissue, and that this reduction leads to suppression of Tor signaling, independently of HIF-1a/Sima. This is consistent with a requirement of Hph for cell growth[49]. In other contexts, Sima is known to regulate Tor-pathway activity via the protein Scylla/REDD1[58,59]. However, this pathway does not appear to be responsible for the effects of hypoxia on Tor activity observed here, as *sima* mutation does not block hypoxia- or starvation-induced Tor suppression. Likewise, Tor suppression is not necessary for the systemic growth reduction induced by hypoxia. Our data suggest that Hph is involved in AA sensing, in addition to its well-described role in oxygen sensing, and that HIF-1a is not involved in this process. Together, this suggests that AA and oxygen sensing converge through Hph in the fat body to modulate systemic growth in response to environmental conditions.

Many of the known *Drosophila* adipokines that affect insulin secretion from the IPCs are regulated by Tor-pathway activity in the fat body, including CCHa-2[10], Egr[17], FIT[11], GBP1 and GBP2[12], and Sun[16]. Our finding that AA availability regulates Hph activity, and that Hph activity modulates Tor signaling, thus places Hph upstream of these known factors, in addition to the separate Sima-dependent and Tor-independent humoral factor(s) that modulate insulin secretion under hypoxia. Several routes by which AA availability regulates Tor have been investigated[49] and Hph may modulate some of these and not others, thereby allowing for different responses to AA starvation and hypoxia. Indeed, our work shows that HIF-1a/Sima is required for the growth-suppressive effect of hypoxia, but not for growth responses to varied dietary AA input, although Hph is involved in both. The mechanisms by which Hph activity, which simultaneously requires AAs and oxygen, allows or promotes Tor signaling is an interesting topic to investigate in future studies, as is the identity of the Tor-independent humoral factor(s) downstream of Sima.

Our results show that hypoxia or loss of fat-body Hph activity Sima dependently represses *Dilp3* and *Dilp5* transcription, while having little or no suppressive effect on *Dilp2* expression. This suggests that specific transcriptional regulation of *Dilp3* and *Dilp5* is an important component of the response to hypoxia. Consistent with this observation, previous studies have shown that the transcription of *Dilp2*, *-3*, and *-5* are independently regulated[7]. Nutrient deprivation reduces expression of *Dilp3* and *-5*, while having no effect on *Dilp2* expression, similar to the effects of exposure to hypoxia. This is consistent with our finding that both AA deprivation and hypoxia suppress Hph activity, although the downstream pathways involved appear likely to be different, as at least some aspects of nutrient deprivation are relayed through the Tor pathway, whereas the hypoxia-specific signal(s) shown

here is not. This observed transcriptional response could conceivably arise secondarily to DILP-release inhibition via autocrine feedback regulation that operates in both *Drosophila* and mammals[60,61]. However, lower secretion of DILPs, which we observe under hypoxic conditions, generally feeds back to induce an increase in the expression of *Dilp3* and *-5* rather than the decrease that we observe. Therefore, the hypoxia-induced alterations of *Dilp3* and *-5* expression appear to be specific transcriptional responses rather than feedback effects. Thus, beyond the identity of the fat-body factor involved, the mechanisms operating in the IPCs by which it regulates insulin-like gene expression and peptide release will be important to study in future experiments.

In mammals, several adipokines regulate β-cell function and insulin secretion, including Leptin, which conveys information about fat storage and is a functional analog of the *Drosophila* fat-derived cytokine Unpaired-2[9,62]. Interestingly, we observed strong increases in fat-body LD size induced by Tor inhibition downstream of AA starvation, hypoxia, or *Hph* mutation, indicating a change in lipid metabolism within the fat tissue. Although this phenotype is Tor-dependent and not upstream of the particular HIF-1a-dependent factor described above, it is possible that additional signals related to lipid metabolism may be released by the fat body in response to hypoxia or starvation, such as a lipid-binding protein or even a lipid per se. For example, the mammalian fatty acid-binding protein 4 is an insulin-modulating adipokine that is influenced by obesogenic conditions that lead to adipose tissue hypoxia[63], and orthologous proteins are encoded by the *Drosophila* genome.

Most organisms stop growing after reaching a genetically predetermined species-characteristic size. Although insight from genetic studies in *Drosophila*[1,6,12,16,17,55] into the mechanisms that regulate body growth with regard to nutrition helps to explain how organisms modulate their growth rate according to nutritional conditions, a mechanism that allows organisms to assess their size and stop their growth when they have reached an optimum has remained elusive. However, recent evidence suggests that body size in insects may be determined by a mechanism that involves oxygen sensing[3], and oxygen availability is known to place limits on insect body size[18,20–22]. According to this recent insight, the limited growth ability of the tracheal system during development may limit overall body size via downstream oxygen sensing[3]. The size of the tracheal system is established at the beginning of each developmental stage and remains largely fixed, aside from terminal branching, as the body grows until it eventually reaches the limit of the system's ability to deliver oxygen. This allows the body to assess its size by sensing internal oxygen concentrations and to terminate growth at a characteristic size that is determined by the size of the tracheal system. Our RNAi screen shows that the FGF receptor Btl, which is a key factor essential for tracheal growth during development, is a main determinant of body size—indeed, *btl* was a stronger hit than known size-governing genes—and our data therefore support the notion that the tracheal system and oxygen sensing may be part of a size-assessment mechanism.

Oxygen homeostasis also requires the coordination of growth between the tissues that consume oxygen and those that deliver it. The development of the oxygen delivery system is therefore oxygen sensitive in both mammals and *Drosophila*. In mammals, local tissue hypoxia promotes angiogenesis via induction of many pro-angiogenic factors, including FGF[64]. In *Drosophila*, tissue hypoxia induces expression of the FGF-like ligand Bnl, leading to branching of the tracheal airway tubes toward oxygen-deficient areas[29,65]. Our study shows that this mechanism operates independently of insulin, as reduced insulin signaling in the trachea has no effect on overall body growth. This system therefore allows an adaptive

response to low oxygen by reducing overall body growth via suppression of insulin signaling, while promoting hypoxia-induced FGF-dependent tracheal growth to increase oxygen delivery.

Cell and tissue hypoxia are also observed in human conditions of obesity and cancer. The insect fat body performs the functions of mammalian fat and liver tissues. Accordingly, perturbation of systemic insulin signaling by adipose and hepatic tissue hypoxia is also observed in mammalian systems[66]. In mammals, obesity induces hypoxia within adipose tissue due to the rarefaction of vascularization of this tissue[67], leading to the release of inflammatory mediators and other adipokines that are associated with the pathophysiology of obesity-related metabolic disorders including diabetes[68–73]. Although loss of normal β-cell activity is considered a main factor in diabetes, the mechanism by which tissue hypoxia affects insulin secretion is poorly understood. Our finding of one or more hypoxia-induced fat-body-derived insulinostatic factors may lead to insights into the role of adipose-tissue hypoxia in obesity and its impact on diabetes. Furthermore, we show a link between oxygen and AA availability in the adipose tissue through Hph-dependent regulation of the Tor pathway, linking these pathways in a common metabolic response to oxygen limitation and nutrient scarcity.

Obesity also causes physical and hormonal changes that affect breathing patterns, leading to apnea and thus intermittent episodes of systemic hypoxia[74]. These hypoxic periods can induce changes in the liver, leading to fatty liver disease and dyslipidemia. The alterations to fat-body lipid metabolism observed here may thus be relevant to human health as well. Furthermore, hypoxia-induced programs play important roles in tumor formation. During cancer development, tumor cells undergo a metabolic reprogramming, the so-called Warburg effect, in which their metabolism shifts from oxidative phosphorylation to glycolysis[75], and activation of HIF-1a is believed to play a key role in this shift. As the hypoxia-sensing mechanism and the insulin-signaling system are conserved between flies and mammals, understanding the effects of hypoxia on the fat body could thus provide insight into many human disease states. It will be of interest to study whether tissue hypoxia also inhibits Tor-pathway activity in mammalian adipocytes.

In conclusion, our study unravels a mechanism that allows organisms to adapt their metabolism and growth to environments with low oxygen. Hypoxia activates a fat-tissue oxygen sensor that remotely controls the secretion of insulin from the brain by inter-organ communication. This involves the inhibition of Hph activity, leading to the activation of a HIF-1a-dependent genetic program within the fat tissue, which then secretes one or more humoral signals that alter insulin-gene expression and repress insulin secretion, thereby slowing growth. We also show that AA scarcity, like oxygen deficiency, inhibits Hph activity, and that the activity of Hph, but not of HIF-1a, is required for Tor activity in the fat body. Thus, in addition to its role in regulating the as-yet unidentified fat-body hypoxia signal via HIF-1a, Hph connects both oxygen and AA levels to the Tor pathway through an unknown HIF-1a-independent mechanism. Given the conservation of oxygen-sensing and growth-regulatory systems, and the influence of oxygen on growth between *Drosophila* and mammals, a similar adaption response may operate in mammals via adipose tissue oxygen sensing to maintain homeostasis.

## Methods

**Fly strains and husbandry.** Unless specified otherwise, animals were reared on standard cornmeal-yeast medium (Nutri-Fly, Bloomington recipe) at 25 °C and 60% relative humidity, under a 12L:12D light cycle. The 1× yeast diet contains 34 g baker's yeast, 6 g agar, 60 g sucrose, 84 g cornmeal, 4.8 mL propionic acid, and 1.6 g Tegosept/Nipagin antifungal agent (in 16 mL 100% ethanol) per liter. Restricted (0.1× and 0.3×) diets contain 3.4 g/L or 10 g/L yeast, respectively, with all other components held constant. Sugar-only diet contains no cornmeal or yeast with

other components held constant. The *AkhR-GAL4::p65* transgene containing the enhanced activator *GAL4::p65*[76] drives very strongly in the larval fat body. The *GAL4::p65* was recombineered into genomic bacterial artificial chromosome construct CH322-147J17[77] obtained from Children's Hospital Oakland Research Institute (Oakland, CA, USA), replacing the first coding exon, and the construct was integrated into the *attP2* genomic site by standard injection methods. The following stocks were obtained from the Bloomington *Drosophila* stock center: *arm-GAL4* (#1560), *btl-GAL4* (#41803), *Cg-GAL4* (#7011), *da-GAL4* (#55850), *elav-GAL4* (#458), *Mef2-GAL4* (#27390), *ppl-GAL4* (#58768), *R96A08-GAL4*[78] (#48030), *repo-GAL4* (#7415), *sima^{KG07607}* (*sima* null)[56] (#14640), *tGPH* insulin-activity sensor[53] (#8164), *UAS-Cas9.P2* (#58986), *UAS-CCHa2-RNAi* (#57183), *UAS-CCHa2-R-RNAi #2* (#25855), *UAS-eiger-RNAi* (#58993), *UAS-InR-RNAi* (#992), *UAS-methuselah-RNAi #1* (#27495) and *#2* (#36823), *UAS-methuselah-like-10-RNAi #1* (#62315) and *#2* (#51753), *UAS-sima-RNAi* (#26207), *UAS-Tor^{TED}* (Tor dominant negative; #7013), and *UAS-Tsc1/2* and *UAS-VALIUM10-Luciferase* (#35789). Vienna *Drosophila* Resource Center (VDRC) RNAi lines used for follow-up studies include *UAS-Akt-RNAi* (#103703), *UAS-bnl-RNAi* (#101377), *UAS-btl-RNA^{GD950}* (#950), *UAS-btl-RNAi^{GD27108}* (#27108), *UAS-CCHa2-R-RNAi #1* (#100290)[10], *UAS-grnd-RNAi #1* (#104538)[17] and *#2* (#43454)[17], *UAS-sima-RNAi* (#106187), *UAS-daw-RNAi* (#105309), and *UAS-Stunted-RNAi* (#23685)[16]. *Hph^{Δ9}* (*fatiga^{Δ9}*, *fga^{Δ9}*)[56] was a kind gift of P. Wappner (Instituto Leloir), *ubi-GFP::ODD* hypoxia sensor (more rigorously, Hph-activity sensor)[54] was a kind gift of S. Luschnig (U. Münster), and *UAS-FLAG::Dilp2*[79] and *UAS-Rheb^{EP50.084,w-}/TM6B, Tb*[80] were gifts of E. Hafen and H. Stocker (ETH Zürich). *UAS-DILP2HF*[51] was a kind gift from S. Park and S. Kim (Stanford). The common laboratory stock *w^{1118}* from VDRC (#60000) was used as a control in many experiments.

**RNAi screen for pupal size.** A list of genes encoding the secretome (secreted proteins) and receptome (membrane-associated proteins) was generated using ENSEMBL and GLAD gene ontology databases. UAS-RNAi lines against 1845 genes for the screening of the secretome and receptome were obtained from the Bloomington *Drosophila* stock center[32] and the VDRC[31], and transformant IDs are given in Supplementary Data 1. Males from each RNAi line were crossed to *da-GAL4* virgin females, animals were allowed to lay eggs for 24 h, and the average size of pupal offspring was determined from images acquired with a Point Grey Grasshopper3 camera using a custom script in MATLAB (The MathWorks, Inc., Natick, Massachusetts, USA)[81]. Each line was given a Z-score, calculated as the number of SD between that line's average and the average of all lines. Lines with a Z-score between −2 and 2, corresponding in this data set roughly to a ±15% size abnormality, were not analyzed further, whereas 89 lines with greater effect were identified.

**Generation of tissue-specific CRISPR lines.** To generate lines for tissue-specific CRISPR-based deletion of *Hph* and *sima/HIF-1a*, double guide RNAs were cloned into plasmid *pCFD6*[82,83] (Addgene #73915). The *Hph* construct should lead to the deletion of part of the genomic region encoding the dioxygenase domain common to all Hph isoforms, and the *sima* gRNAs target the region encoding that protein's DNA-binding domain. The Cas9 target sequences below were synthesized as part of long oligonucleotides that were used to amplify gRNA structural sequences from *pCFD6* using Q5 polymerase (New England Biolabs, #M0491S); plasmid template DNA was then removed by DpnI digestion (NEB, #R0176S). *pCFD6* was linearized by BbsI digestion (NEB, #R3539S) and treated with alkaline phosphatase (NEB, #M0290S) to prevent self-ligation. The PCR products were recombined into the linearized, phosphatase-treated vector by Gibson assembly using the NEBuilder HiFi DNA Assembly Master Mix (NEB, #E2621S). Clones were analyzed by PCR and sequence confirmed. *pCFD6-UAS-Hph^{KO}* and *pCFD6-UAS-sima^{KO}* were integrated into landing site *attP2* (third chromosome) in-house and by BestGene (Chino Hills, CA) using standard methods. To remove *fatiga* or *sima* function in the fat body, two CRISPR-driver stocks—*ppl-GAL4; UAS-Cas9.P2/TM6B, Tb* and *Cg-GAL4; UAS-Cas9.P2/TM6B, Tb*—were made, and these were crossed to the *UAS-gRNA* lines generated above, or to *w^{1118}* as a driver control. The Cas9.P2 variant used here is somewhat weak, so to increase Cas9 expression by increasing GAL4 activity, we incubated the offspring at 30 °C. *Hph* target sites: 5′-GTATCGTGTTAACACGATGA-3′ and 5′-GAATATCAACTGGGATGCGC-3′; *sima* target sites: 5′-GGCTAGATGTCGTCGCTCCA-3′ and 5′-GGCTA-GATGTCGTCGCTCCA-3′.

**Tracheal measurements and hypoxic survival rates.** To determine tracheal length and branching, L3 larvae [96 h after egg lay (AEL) for control; 120 h for *btl-RNAi* animals to account for slowed growth] were heat-fixed in a drop of glycerol on a microscope slide on a 60 °C heat block until they became relaxed. Tracheal cells and branching of the fat-body terminal branches were visualized in whole larvae by imaging of tracheae labeled with GFP or by direct light microscopy. Length and branching were manually quantified using the FIJI (NIH) software package. To determine survival rates of *btl* knockdown animals, timed egg lays were performed, and first-instar (L1) larvae were transferred to fly vials containing standard food and incubated in either hypoxia (5% $O_2$) or normoxia (21% $O_2$, normal atmospheric oxygen concentration), and the number of animals surviving to the pupal stage was determined. The low-oxygen environment was generated by

slowly passing 5% $O_2$ + 95% $N_2$ through a clear, airtight plastic chamber placed inside a 25 °C, 12L:12D incubator. Gas was bubbled through water upstream and downstream of the chamber, to provide humidity and to prevent infiltration of outside air, and vented outside the incubator. The chamber was flushed with high-flow-rate gas for 5 min after any opening.

**Analysis of developmental timing and growth rates**. To synchronize larvae for growth rate and developmental-timing assays, flies were allowed to lay eggs for 4 h on apple-juice agar plates supplemented with yeast paste. Newly hatched L1 larvae were transferred into vials with standard food (30 per vial to prevent crowding) 24 h later. To determine growth rates, synchronized larvae were weighed at time points 75–96 h AEL in groups of 5–10 animals. For developmental-timing experiments, pupariation time was determined manually.

**Ex-vivo organ co-culture**. For organ co-culture experiments, brains and fat bodies were dissected from 96 h AEL wild-type $w^{1118}$ larvae or 120 h AEL *Hph*-mutant larvae (to adjust for their slow growth) in cold Schneider's culture medium (Sigma-Aldrich #S0146) containing 5% fetal bovine serum (Sigma). Fifty brains, with or without fat bodies present, were incubated in 50 μL culture medium for 16 h at 25 °C in small dishes in moist chambers in either hypoxia or normoxia. After incubation, brains were removed for immunostaining and quantitative real-time PCR (qPCR) analyses.

**Expression analysis by qPCR**. To quantify gene expression by qPCR, mRNA was prepared from samples of five whole larvae each (96 h AEL for *btl>* + controls and normoxia animals, and 120 h AEL for *btl-RNAi* larvae, animals raised under hypoxia, and *Hph* mutants to adjust for their slower growth) using the RNeasy Mini Kit (Qiagen #74106) with DNase treatment (Qiagen #79254). RNA yield was determined using a NanoDrop spectrophotometer (ThermoFisher Scientific), and cDNA was generated using the High-Capacity cDNA Reverse Transcription Kit with RNase Inhibitor (ThermoFisher #4368814). qPCR was performed using the QuantiTect SYBR Green PCR Kit (Fisher Scientific #204145) and an Mx3005P qPCR System (Agilent Technologies). Expression levels were normalized against *RpL32*. FlyPrimerBank pre-computed oligo pairs were used for some target genes. The primer pairs used are given in Supplementary Table 1.

**Western blotting analysis**. To quantify signaling pathway activity downstream of InR, the level of Akt phosphorylation (pAkt) was determined by western blotting. For hypoxia experiments, larvae were transferred to 5% $O_2$ at 72 h AEL and analyzed 24 h later, compared with 96 h AEL normoxic controls. *Hph*-mutant larvae were assayed at 120 h AEL. For each sample, three larvae were homogenized in SDS loading buffer (Bio-Rad), boiled for 5 min, centrifuged at $14,000 \times g$ for 5 min to pellet debris, and electrophoresed through a precast 4–20% poly-acrylamide gradient gel (Bio-Rad). Proteins were transferred to a polyvinylidene difluoride membrane (Millipore), and the membrane was blocked with Odyssey Blocking Buffer (LI-COR) and incubated with rabbit anti-pAkt (Cell Signaling Technology #4054, diluted 1:1,000) or rabbit anti-pan-Akt (Cell Signaling Technology #4691, diluted 1:1,000), and mouse α-Tubulin (Sigma #T9026, diluted 1:5,000) in Odyssey Blocking Buffer (LI-COR) containing 0.2% Tween 20. Primary antibodies were detected with goat secondary antibodies—IRDye 680RD anti-mouse (LI-COR #925-68070) and IRDye 800CW anti-rabbit (LI-COR #925-32210) diluted 1:10,000—and bands were visualized using an Odyssey Fc imaging system (LI-COR). Uncropped blots are presented in the Source Data file.

**ELISA of circulating tagged DILP2**. *Dilp2 > DILP2HF*[51] animals were reared under normal conditions until 24 h before wandering and then transferred to starvation medium (1% agar in water) in normoxia, to normal food under hypoxia (5% $O_2$), or back to normoxia and normal diet for 12 h. Hemolymph was extracted by cutting the larval cuticle and pipetting the released droplet into a collection vial. Collected hemolymph was heat-treated at 60 °C for 5 min, centrifuged to remove any debris or aggregates, and used in an ELISA against the HA and FLAG tags. F8 MaxiSorp Nunc-Immuno modules (Thermo Scientific #468667) were incubated at 4 °C overnight with 5 μg/mL anti-FLAG (Sigma-Aldrich #F1804) in 200 mM $NaHCO_3$ buffer (pH 9.4). After two washes with phosphate-buffered saline (PBS) + 0.1% Triton X-100 (PBST), the plate was blocked with PBST + 4% non-fat dry milk for 2 h at room temperature and washed again three times in PBST. One microliter of hemolymph or synthetic HA::spacer::FLAG peptide standard (DYKDDDDKGGGGSYPYDVPDYamide) was diluted into 50 μL PBST + 25 ng/mL mouse anti-HA peroxidase (Roche 12013819001) + 1% non-fat dry milk, added to the wells, and incubated overnight at 4 °C. Fluid was removed and wells were washed in PBST six times. One-step Ultra TMB ELISA substrate (Thermo Scientific #34028) was added to each well (100 μL) and incubated for 15 min at room temperature. Sulfuric acid (2 M, 100 μL) was added to stop the reaction and absorbance at 450 nm was measured using a plate reader. Plate, peptide standard, *UAS-DILP2HF*, anti-FLAG, anti-HA, substrate, and detailed protocol were all generously provided by S. Park and S. Kim (Stanford University).

**Immunostaining**. Brains were removed from ex-vivo cultures or dissected from larvae 96 h AEL (normoxia) or 120 h AEL (hypoxia and *Hph* mutants) in cold Schneider's culture medium and fixed in fresh 4% paraformaldehyde in PBS for 1 h. Tissues were washed four times in PBST (PBS containing 0.1% Triton X-100), blocked with PBST containing 3% normal goat serum (Sigma-Aldrich #G9023) at room temperature for 1 h, and incubated with primary antibodies overnight at 4 °C. The following primary antibodies were used: rat anti-DILP2[52] (kind gift of P. Léopold, U. Nice) at 1:250 dilution; mouse anti-DILP3 at 1:200 dilution (kind gift of J. Veenstra, U. Bordeaux); rabbit anti-DILP5[84] at 1:2,000 dilution (kind gift of D. Nässel, U. Stockholm); rabbit anti-phospho-ribosomal pS6 (anti-pS6)[57] at 1:200 (kindly given by A. Teleman, DKFZ). Tissues were washed three times in PBST and incubated in secondary antibodies: goat anti-rabbit Alexa Fluor 488 (Thermo Fisher Scientific, #A32731), goat anti-rat Alexa Fluor 555 (Thermo Fisher Scientific, #A21434), and goat anti-mouse Alexa Fluor 647 (Thermo Fisher Scientific, #A28181), all diluted 1:500 in PBST, overnight at 4 °C. After four washes in PBST, tissues were mounted in Vectashield (Vector Labs #H-1000) and imaged using a Zeiss LSM 800 confocal microscope and Zen software. Retained insulin levels were calculated using the FIJI software package.

**Fat-body imaging**. To measure in-vivo intracellular insulin-signaling transduction in the fat body, we made use of a GFP-Pleckstrin Homology domain protein fusion protein (GPH) expressed from the ubiquitously active *β-Tubulin* promoter (tGPH). Insulin-signaling activity causes recruitment of the GPH protein to the plasma membrane[53]. Hypoxia was induced at 72 h AEL, and hypoxic animals were assayed for fat-body GFP localization 24 h later together with age-matched 96 h AEL normoxic controls. Fat bodies were dissected in cold Schneider's medium and immediately imaged. To determine whether the fat body experiences biologically relevant hypoxia under 5% ambient oxygen levels and whether Hph activity is responsive to AA levels, we used a GFP::ODD sensor line, made up of the ODD of Sima fused to GFP, expressed throughout the animal from the *ubiquitin-69E* promoter[54]. Similar to endogenous Sima protein, the GFP::ODD reporter is degraded at a rate reflecting Hph activity; thus, in normal conditions, the GFP signal is low, whereas in hypoxia or other conditions of Hph inhibition, the GFP signal is increased. Unmodified RFP expressed from the same regulatory elements provides a ratiometric denominator. For hypoxia tests, larvae were reared under normoxia or hypoxia until 96 or 120 h AEL, respectively, at which point fat bodies were dissected in cold Schneider's medium and immediately imaged. For protein-starvation assays, animals were reared in normoxia at 25 °C on standard (1× yeast) diets until 92 h AEL and then transferred either to a fresh vial of the same diet or to protein-starvation medium (sugar alone) for 4 h; fat bodies were dissected in cold Schneider's medium (non-starved animals) or cold PBS (starved animals, to prevent uptake of AAs). Confocal stacks of endogenous xFP fluorescence were taken immediately. Levels of GFP and RFP signals were integrated using the FIJI package, and the ratio between them was calculated. LDs were imaged using label-free Coherent Anti-Stokes Raman Scattering (CARS) microscopy. For normoxia assays, animals were reared on standard diet in normoxia until harvest (for $Hph^{\Delta 9}$, at 120 h; otherwise, 96 h). For hypoxia assays, $w^{1118}$ and $sima^{KG07607}$ animals were reared for 120 h in hypoxia; $Hph^{\Delta 9}$ animals died when reared in hypoxia, so these animals were reared in normoxia for 120 h and then transferred to hypoxia for 16 h before harvest. For measurements of response to AA starvation, $w^{1118}$ and $sima^{KG07607}$ animals were reared on standard (1× yeast) for 96 h and $Hph^{\Delta 9}$ for 120 h, and then transferred to sugar-only diet for 16 h. Fat bodies were dissected in cold Schneider's culture medium and immediately imaged using a Leica TCS SP8 confocal microscope[85,86], using a picoEmerald laser and BS635 filter for lipids along with these settings: IR power, 2000 mW; OPO, 2000 mW at 780–940 nm, pulse width 5–6 ps, 80 MHz; pump, 1174 mW at 1064 nm, pulse length 7 ps, 80 MHz. To image lipids, the lasers were tuned to the C–H stretching vibration: pump beam, 816.4 nm; Stokes beam, 1,064.6 nm; laser output, 1.3 W; scan speed, 400 Hz. Signals were collected from the epi-CARS and epi-SHG detectors. LD size was quantified using the FIJI software package.

**RNAi mini-screen of fat-body factors and IPC receptors**. *R96A08-GAL4*, targeting the IPCs, was crossed to RNAi constructs against receptors, and the fat-body driver *AkhR-GAL4::p65* was crossed to lines targeting secreted factors. Fat-body driver *ppl-GAL4* was crossed to *daw-RNAi* and *UAS-Rheb*. Newly hatched larvae from 4 h egg lays were transferred at 24 h AEL to standard food vials and incubated at 25 °C in normoxia or hypoxia (5% $O_2$) until pupariation. Pupal size was measured, and for each genotype the ratio for each possible hypoxic and normoxic vial pair was computed to make up each data set. Experimental data were compared with data from controls expressing *UAS-Luciferase* in a background matching that of the RNAi lines, except for the *ΔGbp1,2* deletion line, which was normalized to $w^{1118}$, and *ppl>* crosses, which were compared with *ppl-GAL4/+*.

**Statistical analyses**. Statistical tests and *P*-values are given within each figure legend. Statistics were calculated using GraphPad Prism Software, applying analysis of variance, Dunnett's test, or Kruskal–Wallis non-parametric tests for multiple-comparisons tests and two-tailed Student's *t*-tests or two-tailed exact Mann–Whitney non-parametric tests for pairwise comparisons. Error bars indicate

SEM and *P*-values are indicated as follows: $*P < 0.05$, $**P < 0.01$, $***P < 0.001$, and $****P < 0.0001$.

**Reporting summary**. Further information on experimental design is available in the *Nature Research* Reporting Summary linked to this article.

## Data availability

Data that support the findings of the current study are available from the corresponding author on reasonable request.

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

## Acknowledgements

We thank our Erasmus-program visiting undergraduate Niyati Jain for help. We are grateful to Seung Kim, Pierre Léopold, Stefan Luschnig, Dick Nässel, Sangbin Park, Aurelio Teleman, Jan Veenstra, and Pablo Wappner for kindly sharing antibodies, fly lines, and reagents. We also thank Addgene, the Bloomington *Drosophila* Stock Center, and the Vienna *Drosophila* Resource Center for making plasmids and fly lines broadly available. This work was supported by the Novo Nordisk Foundation STAR grant and an Innovation foundation grant to J.L.H. and K.F.R. and by Novo Nordisk Foundation grant 16OC0021270 and Danish Council for Independent Research Natural Sciences grant 4181-00270 to K.F.R. K.A.H. was supported by Villum Foundation grant 15365 to K.A.H.

## Author contributions

M.J.T., A.F.J., C.F.C., T.K., A.M., D.K.S., D.F.M.M., E.T.D., S.K.P., K.A.H. and K.F.R. performed the experiments and analyzed the data. M.J.T., A.F.J., C.F.C., J.L.H., K.A.H. and K.F.R. designed the experiments. M.J.T., K.A.H. and K.F.R. wrote the manuscript.

## Additional information

**Competing interests:** J.L.H. is an employee and shareholder of Novo Nordisk A/S. A.F.J. is an employee of Novo Nordisk A/S. All remaining authors declare no competing interests.

