## [Peer Review File · Nature Communications]

Reviewers' comments:

Reviewer #1 (Remarks to the Author):

Organismal growth is meticulously controlled by integrating multiple internal and external cues such as organ size and nutrient availability. One of the external cues that have been shown to affect growth, particularly in invertebrates, is oxygen availability. Understanding the mechanistic details of how oxygen availability is sensed and subsequently tethered to mechanisms controlling growth of the organism has significant biological relevance.

The authors addressed the broader question of how organismal growth is regulated at a genetic level. They performed a screen to identify genetic regulators of growth by knocking down 1845 genes (encoding the *Drosophila* secretome and receptome) in the whole larvae and measuring their effect on the size of the pupae. The screen revealed a number of known regulators of growth and developmental timing validating the approach. Additionally, it indicated that the *Drosophila* FGF receptor ortholog *breathless* (*btl*) could be a positive regulator of organismal growth since loss of *btl* lead to smaller pupae.

Since *btl* signaling is a known regulator of tracheal branching in *Drosophila*, they further investigated if loss of growth in *pan-larval>btl-RNAi* animals is due to hypoxia induced by reduced tracheal branching.

Taking advantage of a number of elegant genetic approaches they show that. 1) *pan-larval>btl-RNAi* reduces tracheal branching in the larvae, 2) loss of tracheal branching causes hypoxia in the animals, 3) when the fatbody senses hypoxia through the conserved *Hph->Sima* pathway, it secretes an unknown factor that remotely inhibits insulin release from the larval insulin producing cells. 4) it is this loss of insulin release that is responsible for hypoxia induced inhibition of growth in the larvae.

The data seems quite convincing. There are a few concerns (listed below) that should be addressed to make the study more convincing.

My major concern however is that the work does not add much new information to the already existing body of work. Indeed, previous studies have shown:

- 1) *btl* is a well know regulator of tracheal branching.
- 2) Loss of tracheal branching will cause hypoxia and hypoxia is a well-known regulator of larval growth.
- 3) The fact that hypoxia is being sensed in the FB and regulates growth by inhibiting insulin release from the IPCs is very interesting. However, the findings have already been reported in another paper from 2014 (Wong et al. *PlosOne* 2014). While this paper does the experiments more exhaustively and does impart more confidence in the old finding, they don't use any new technique or approach for validation.

Other concerns and improvements:

- 1) Representation of the tracheal branching phenotype is not convincing.

Figure 4 A: Authors report that the loss of *btl* leads to decreased tracheal branching in the larval FB. While this is the expected outcome, the image provided does not convincingly prove the point. The authors do not show any spatial information regarding which region of the FB we are looking at. This makes it difficult to judge whether the differences are real or an artifact of observing non-identical sites. Terminal tracheal branching can be extremely unpredictable and unless comparisons are made at an exact defined location where the number of branches can be predictably counted it can be very difficult to interpret the data. Please provide a better image, preferably showing the same section of the FB between the experimental and control flies to make

the image more convincing. Alternatively, larval whole mounts with GFP tagged trachea could be used to show that the experimental animals have lower tracheal branching.

2) To identify the tissue in which *btl* is required for larval growth the authors use *btl-Gal4* (trachea), *elav-Gal4* (neurons), *mef2-Gal4* (muscle) and *ppl-Gal4* (FB). They also mention that *btl* is expressed in glial cells along with the tracheal cells. It would be nice to rule out the involvement of the glia in the *btl* dependent growth phenotype by using a glial specific driver like *Repo-gal4*.

3) Figure 6e: The authors claim that *DILP2/3/5* expression does not change between normoxic vs hypoxic conditions when larval brains are cultured alone. However, when brains are co-cultured with FB cells *DILP2/3/5* expression is significantly suppressed in the hypoxic condition compared to normoxic conditions. Based on this data they conclude that the FB sends an inhibitory signal to the brain.

This conclusion is valid only if *DILP2/3/5* expression does not change between the brain and brain+FB samples under normoxic conditions.

If *DILP2/3/5* expression tends to go up when brain is co-cultured with FB. Then the reduction in expression in response to hypoxia could be due to inhibition of a positive signal that is transmitted from the FB to the brain. Thereby completely changing the conclusion drawn by the authors.

The authors do not show that data. It will be best if they simply show expression of *DILPs* relative to the reference gene under all conditions instead of reporting hypoxia/normoxia ratios alone. It will help the readers interpret the data better.

4) The authors go through a number of ligands that are known to regulate *DILP* release. They however missed to check *Daw* as a candidate. *Daw* has been shown by at least two studies to regulate *DILP* release and is also released from the FB (PMID: 24706779 and 24244197). It would be worth checking if *Daw* expression in the FB changes in response to hypoxia and if it plays a role in the phenotypes presented.

Reviewer #2 (Remarks to the Author):

The authors have studied the relation between O₂, insulin, insulin-producing cells in brain and fat bodies. The results are interesting and the study is well done by and large.

1. why is there lethality with ubiquitous expression of *Dilp2* with daughterless *Gal4*?
2. Is insulin downstream of the ecdysone system?
3. How do the *btl-RNAi* pupae pupae die? Not clear as to what hypoxia-induced damage is here? do the authors have any evidence?
4. How do the authors explain that the fat droplets decrease in size and quantity in the fat bodies under hypoxia in *btl*-deficient flies?

Response to the reviewers

Reviewer #1

The reviewer finds that our work takes advantage of a number of elegant genetic approaches to address the broad question of how organismal growth is regulated. Our study provides mechanistic insight into how oxygen availability is sensed and coupled to growth of the organism and the reviewer points out that this is of significant biological relevance. We thank the reviewer for this and for very insightful comments. Although the reviewer finds our study convincing and rigorously performed, the reviewer raises some concerns and suggests that it would benefit from additional novel findings. We have now addressed, either by changing imprecise wording or performing clarifying experiments, all of the reviewer's individual comments, and we also have made some very exciting new discoveries that add great novelty to our work. I will discuss this component first to address the reviewer's main point and then the individual comments.

Hypoxia (low oxygen level) is known to slow growth, leading to delayed development and smaller adults in most, if not all, organisms, yet the underlying mechanism has remained unknown. Our study *provides the first model for explaining the mechanism underlying the general hypoxia-adaptation response to reduce growth*. Furthermore, our study is *the first to show that fat tissue is a central sensor of internal oxygen availability that modulates systemic growth through the release of one or more secreted factors acting directly on the brain to regulate insulin release*. This is an entirely new concept that the fat tissue acts as the primary internal sensor of oxygen availability. Finally, we now show in the revised manuscript that previously described *nutrient-sensing mechanisms within the fat merge with the oxygen-sensing system through a novel Hph-dependent mechanism that is required for activation of the Tor pathway*, the main cellular nutrient-response signaling pathway. In the course of experiments for the revision of our work, we discovered a new function of HIF-1 α Prolyl Hydroxylase (Hph) in linking oxygen- and nutrient-sensing through Target of rapamycin (Tor), which will be of great importance to fields ranging from metabolic research to cancer biology and development and which adds exciting novelty to our findings.

Reviewer #1 point 1-3:

My major concern however is that the work does not add much new information to the already existing body of work. Indeed, previous studies have shown:

- 1) btl is a well-known regulator of tracheal branching.*
- 2) Loss of tracheal branching will cause hypoxia, and hypoxia is a well-known regulator of larval growth.*
- 3) The fact that hypoxia is being sensed in the FB and regulates growth by inhibiting insulin release from the IPCs is very interesting. However, the findings have already been reported in another paper from 2014 (Wong et al. PlosOne 2014). While this paper does the experiments more exhaustively and does impart more confidence in the old finding, they don't use any new technique or approach for validation.*

Author response to points 1 and 2:

The reviewer argues that the work does not add much to the state of the art because (1) Breathless (Btl) is a known regulator of tracheal branching and (2) it is appreciated that the loss of tracheal branching will cause hypoxia, which is a well-known regulator of larval growth. While it is true that Btl is known to regulate tracheal branching, we performed a large-scale screen to identify the main mechanisms that determine overall body growth, not tracheal branching. The main hit from that screen that we focused on, *btl*, produced the second-smallest animals we observed – that is, it was a stronger hit than any of the known factors regulating systemic growth, indicating that hypoxia brought about by tracheal limitation is among the most powerful regulators of growth and body size yet observed. This is a novel finding, which is relevant to the reviewer's points 1 and 2. In other words, while other work has linked *btl* to development of the tracheal system *per se*, we now show that it is at the core of perhaps the most important signaling pathway that regulates growth of the entire organism. One of the major unsolved questions in biology is how organisms grow to be their species-specific “correct” body size. This implies the existence of a size-assessment system that monitors organismal size during development and allows organisms to stop their growth once they have reached a certain size, but the underlying mechanism has remained unknown. Our unbiased screening approach that identified *btl* suggests that the oxygen delivery capacity of the tracheal system is limiting for growth of the organism and may be the mechanism by which insects “assess” and regulate their own size, thereby adding insight to a longstanding genetic and ecological question. Thus, a major novelty in our paper is the molecular explanation of body-size determination by oxygen-delivery capacity, which has been a subject of considerable speculation.

Author response to point 3 regarding Wong *et al.*:

The reviewer also finds that aspects of our work are related to work previously reported in Wong *et al.*¹, although the reviewer feels that our studies are more rigorous and thorough. We would therefore like to take the opportunity to explain in detail the major difference and the major conceptual advance that our work represents compared to that of previous reports. Firstly, the work by Wong *et al.* suggested that insulin regulates specific growth of the tracheal system under hypoxia to modulate systemic larval growth. This model is totally different from ours and is in direct tension with broader ecological and physiological findings. Their model therefore does not provide insight that explains the mechanisms by which oxygen availability is sensed and coupled to growth of the entire organism. Our work does that, and this is a novel and significant finding that represents a conceptual advance in a field that has received a lot of attention recently, namely inter-organ communication in regulation of growth and metabolism. In fact our work shows the exact opposite of Wong *et al.* – we find that tracheal growth is insulin-independent, as surely it must be if the tracheae are to grow during oxygen deprivation as an adaptive response to deliver more oxygen. This is broadly supported by literature showing increased tracheal branching under hypoxia. If insulin signaling decreases during hypoxia, this increased tracheal growth cannot be dependent on insulin.

While some work presented by Wong *et al.* agrees with our study by indicating that hypoxia reduces insulin signaling, their study provides no solid evidence that this involves direct communication between the fat tissue and the brain. We show that a hypoxia-induced factor(s) released from the fat body acts directly on the brain, not via any intermediate tissue, through our elegant *ex-vivo* experiments. Their study also provides no direct evidence that HIF-1 α in the fat body inhibits growth systemically in response to hypoxic conditions. We show this in the revised manuscript (see Fig. 8A, 8B and 8C), where we have performed highly efficient tissue-specific CRISPR/Cas9-induced disruption of *HIF-1 α* in the fat body. We now show that this knockout of *HIF-1 α* (also called *sima* in *Drosophila*) partially rescues hypoxia-induced growth inhibition and suppression of insulin production and release. This for the first time directly shows that the fat tissue is mediating this hypoxia-adaptation response to reduce growth, via HIF-1 α -dependent secretion of one or more humoral factors that regulate insulin secretion by inter-organ communication. Furthermore, we support this conclusion by fat-body-specific CRISPR/Cas9-mediated knockout of *HIF-1 α Prolyl Hydroxylase* (*Hph*). The Wong *et al.* study provides no evidence that *Hph* is involved in or regulates growth, while we show that disruption of *Hph* in the fat tissue alone is sufficient to reduce growth and insulin signaling (Fig. 6G, 6H, and 6I). In the revised manuscript, we show that CRISPR-mediated disruption of *Hph* function in the fat body,

which induces a genetic hypoxic response, reduces systemic growth, and blocks transcription and secretion of brain insulin even under normoxia. Thus, Wong *et al.* reach a different conclusion in work that is not focused in the exciting new concept that we present here: *that the fat tissue is a central oxygen sensor that remotely controls systemic growth via insulin.* We therefore believe that the two studies are of a different nature, and that the observations by Wong *et al.* actually strengthen our story rather than weakening it.

We believe that a problem with the Wong *et al.* paper which led them to their (in our view, erroneous) conclusion regarding insulin-dependent tracheal growth is that the experiment in question was based on over- or mis-expression of the *Insulin Receptor (InR)* in the tracheal cells, which is a gain-of-function likely to produce neomorphic effects. In fact, other experiments in their report, like ours, show that loss of insulin signaling in the tracheae has no influence on overall body growth. This shows that insulin has no role in regulating the growth of the tracheal system to influence body size and therefore places insulin “downstream” of hypoxia, as we show. Indeed, regulation of tracheal growth by insulin signaling would be counterproductive – under hypoxic conditions, the tracheae must grow to increase their oxygen-delivery capacity, and because insulin secretion is reduced under this condition, the tracheae should be insulin-insensitive, or at least capable of insulin-independent growth, at least under hypoxia. This is the conclusion that our data supports, and this provides a molecular mechanism explaining decades of ecological, physiological, and genetic observations. Thus, a major novelty in our paper is the explanation of body-size regulation by oxygen-delivery capacity of the tracheal airway system, which has been a subject of considerable speculation. It is well-known that organisms adapt growth to oxygen levels, but *our work and model are the first that explain the exact mechanism underlying the growth adaptation to hypoxia in any animal*, which makes our work a major advance that will be important for several fields ranging from ecology to physiology and growth control in both normal development and diseases such as cancer, in which cells switch to a metabolic state resembling hypoxia.

Author response to point 3 regarding new methods and novelty:

We also thank the reviewer for praising the rigor of our work, which we have even further increased in the revised version of our manuscript by introducing fat-body-specific CRISPR/Cas9-mediated *Hph* and *HIF-1a/sima* gene knockouts as described above. This also addresses the reviewers’ request of the use of new technological approaches. These tools are novel and appear to be much more effective than current RNAi-based ones against *Hph* and *HIF-1a* in *Drosophila*, an issue that has been a major obstacle to tissue-specific loss-of-function studies of these genes. These new transgenic CRISPR-lines can easily be used for tissue-specific knockout of these factors in any tissue and will benefit the *Drosophila* community as we will make them freely available. We have now also directly measured levels of insulin (DILP2) released into the hemolymph, the insect circulatory fluid, by ELISA (see Fig. 5E of the revised manuscript). This shows that the elevated DILP levels observed in the insulin-producing cells (IPCs) under hypoxia do indeed reflect reduced release into the circulation and that hypoxia induces a stronger drop in circulating DILP levels than total starvation. This is an interesting finding that supports our contention that the factor released by the fat body in response to hypoxia is a previously uncharacterized *Hph*- and *HIF-1a*-dependent signal that acts as an inhibitor of insulin secretion. The existence and nature of this insulin-regulatory secreted factor(s) will be of general interest both from a medical perspective and in many other fields of research.

During the revision of our work, we also discovered a new function for *HIF-1a* Prolyl Hydroxylase (*Hph*) as briefly described above, which we believe will be of great importance to fields ranging from metabolic research to cancer biology and development. By rearing wild-type animals and *Hph* and *HIF-1a/sima* mutants on diets containing normal or reduced protein concentrations, in both hypoxia and normoxia, we have shown now that *Hph* is required to regulate growth (Fig. 7A) and Tor activity (Fig. 7B) in response to both oxygen and dietary amino acids. Hypoxia is known to block the marking of *Hph* target proteins for degradation (also shown in Fig. 6B and S5A). We now show that short-term amino-acid starvation under normoxic conditions also leads to decreased degradation of an *Hph*-target reporter, indicating that these nutrients also regulate *Hph* activity (Fig. 7D). Furthermore, whereas in wild-type animals, increased levels of amino acids and oxygen promote Tor activity, in *Hph* mutants, Tor activity is suppressed independently of dietary conditions and oxygen. Thus, *Hph* activity, which is reduced by amino-acid starvation and hypoxia,

is required for Tor activity in the fat body. We found that unlike animals lacking *Hph* function, *HIF-1a/sima* null mutants were still able to respond to dietary protein concentration by adapting their growth (Fig. 7A) and to downregulate Tor activity in response to oxygen or amino-acid deficiency (Fig. 7B). These data indicate that HIF-1a/Sima is not required for growth response to amino acids or for Tor inhibition, which rules out the contribution of the known HIF-1a-REDD1-Tor pathway^{2,3} to this effect. Furthermore, we also found that the insulinostatic effects of fat-body hypoxia (our initial entry into this research) are *Tor-independent* and *HIF-1a/Sima-dependent*. Thus, we have added the novel finding that *Hph* integrates oxygen and amino-acid availability and regulates body growth via a HIF-1a/Sima-dependent signaling pathway in the fat body that alters systemic insulin signaling, while in parallel controlling Tor activity and aspects of downstream lipid metabolism (Fig.7C) through a separate pathway. Amino-acid availability has long been recognized as a regulator of Tor activity, although the mechanisms by which this regulation occurs are not clearly defined. Understanding how *Hph* is involved in sensing nutrients and regulates Tor will be an important topic to investigate in future studies of nutritional integration and growth.

The *Hph*-HIF-1a pathway mediates a metabolic switch known to drive growth of many human cancers, in which tumor cells undergo a metabolic reprogramming, the so-called “Warburg effect”, shifting from oxidative phosphorylation to glycolysis⁴. Activation of HIF-1a is believed to play a key role in driving this shift. *Hph* and its target, the transcription factor HIF-1a, have therefore been extensively studied for their role in the switch in tumor glucose metabolism as well as in the cellular response to oxygen. Interestingly, we discovered that *Hph* also plays a key role in transducing nutrient sensing. Our novel data included in the revised manuscript show that *Hph* is required for nutrient-dependent activity of the Tor pathway, one of the most highly studied signaling pathways, which is often hyperactivated in cancers. In addition to oxygen, organisms and cancer cells require nutrients for growth and development. We now provide a novel and highly significant link between the oxygen-sensing mechanisms and Tor-dependent nutrient sensing that connects two major pathways that regulate growth and metabolism in response to the two key environmental factors necessary for growth. These new findings have major implications for understanding the mechanism by which organism adapt their growth to the environment as well as metabolic disease biology and cancer research. The data are included in an entirely new figure (Fig. 7) as well as additional data in Figure S5 and S6 of the revised manuscript. All together we believe that this clarification as well as our additional new findings further increase novelty of our work and address these points raised by the reviewer.

Other concerns and improvements:

We thank the reviewer for the specific points brought up. They have all been addressed, and we believe the additions have improved the manuscript.

Reviewer point 1:

Representation of the tracheal branching phenotype is not convincing.

*Figure 4 A: Authors report that the loss of *btl* leads to decreased tracheal branching in the larval FB. While this is the expected outcome, the image provided does not convincingly prove the point. The authors do not show any spatial information regarding which region of the FB we are looking at. This makes it difficult to judge whether the differences are real or an artifact of observing non-identical sites. Terminal tracheal branching can be extremely unpredictable and unless comparisons are made at an exact defined location where the number of branches can be predictably counted it can be very difficult to interpret the data. Please provide a better image, preferably showing the same section of the FB between the experimental and control flies to make the image more convincing. Alternatively, larval whole mounts with GFP tagged trachea could be used to show that the experimental animals have lower tracheal branching.*

Author response to point 1:

As this reviewer mentioned, a tracheal-branching phenotype is the expected outcome of *breathless* knockdown, so this is not a controversial point. However, the reviewer requests a broader overview of the tracheal system in Figure 4, in particular focusing on the region we located in each prep for analysis. We have clarified the location at which we measured tracheal phenotypes and included this in the revised Figure

4, and we also made sure that the text reflects that the same anatomy was measured in each instance. These changes improve the figure, and we thank the reviewer.

Reviewer point 2:

*To identify the tissue in which *btl* is required for larval growth, the authors use *btl-Gal4* (trachea), *elav-Gal4* (neurons), *mef2-Gal4* (muscle) and *ppl-Gal4* (FB). They also mention that *btl* is expressed in glial cells along with the tracheal cells. It would be nice to rule out the involvement of the glia in the *btl* dependent growth phenotype by using a glial specific driver like *repo-GAL4*.*

Author response to point 2:

This is a nice point raised by the reviewer, whom we thank for it. We have now tested this by expressing *breathless-RNAi* in glial cells using the *repo-GAL4* driver, as suggested, and we did not see any significant size changes. These data are included in Figure 2C of the revised manuscript.

Reviewer point 3:

*Figure 6e: The authors claim that *DILP2/3/5* expression does not change between normoxic vs hypoxic conditions when larval brains are cultured alone. However, when brains are co-cultured with FB cells *DILP2/3/5* expression is significantly suppressed in the hypoxic condition compared to normoxic conditions. Based on this data they conclude that the FB sends an inhibitory signal to the brain.*

*This conclusion is valid only if *DILP2/3/5* expression does not change between the brain and brain+FB samples under normoxic conditions.*

*If *DILP2/3/5* expression tends to go up when brain is co-cultured with FB. Then the reduction in expression in response to hypoxia could be due to inhibition of a positive signal that is transmitted from the FB to the brain. Thereby completely changing the conclusion drawn by the authors.*

*The authors do not show that data. It will be best if they simply show expression of *DILPs* relative to the reference gene under all conditions instead of reporting hypoxia/normoxia ratios alone. It will help the readers interpret the data better.*

Author response to point 3:

We thank the reviewer for pointing out that the way some data were presented in the original Figure 6 made it unclear whether the fat-body signal is indeed inhibitory. We have now changed our original Fig. 6E (Fig. 6D of the revised manuscript) to make it easier to interpret as suggested by the reviewer. Importantly, we have also reformatted the figure showing the *ex vivo* data of *DILP* retention in the IPCs in response to hypoxia (Fig. 6E of the revised manuscript). This reformatting makes it clear that *DILP2* is retained when brains were co-cultured with wild-type fat-body tissue under hypoxia as compared to brains cultured alone under these conditions or under normoxia. This indicates that *DILP* release is actively inhibited by the factor(s) released by the fat body under hypoxia, rather than reduced by the lack of an activator. On the other hand, co-culturing wild-type brains and fat bodies under normoxia reduced *DILP2* retention, suggesting that the fat body tissue released humoral signals that promote insulin secretion in well-fed animals reared under normal oxygen conditions, consistent with previous reports⁵⁻⁸. We also show now in the revised manuscript that hypoxia on normal diet causes a greater drop in circulating *DILP2* levels than complete starvation (under normoxia) (Fig. 5E), supporting the release of an inhibitory fat body-derived factor under hypoxia.

Reviewer point 4:

*The authors go through a number of ligands that are known to regulate *DILP* release. They however missed to check *Daw* as a candidate. *Daw* has been shown by at least two studies to regulate *DILP* release and is*

also released from the FB (PMID: 24706779 and 24244197). It would be worth checking if *Daw* expression in the FB changes in response to hypoxia and if it plays a role in the phenotypes presented.

Author response to point 4:

We thank the reviewer for bringing this to our attention. We have now tested whether *Dawdle* contributes to this phenotype by expressing RNAi against *dawdle* in the fat body and measuring pupal size in normoxia and hypoxia. RNAi against a factor required for hypoxia-induced DILP retention, we reason, should block or reduce the size decrease seen in hypoxia, but we did not see any such “rescue” effect in our experiments (Fig. S6B). Thus, *Dawdle* does not appear to be required for this effect of hypoxia, and we have now included this information in the revised manuscript and cited the relevant paper that the reviewer suggests.

Reviewer #2

Reviewer #2 points out that our large study is interesting and well done. We thank the reviewer for that comment.

Reviewer point 1:

Why is there lethality with ubiquitous expression of Dilp2 with daughterless Gal4?

Author response to point 1:

This is an interesting question, and we conjecture that lethality arises from hypoglycemia-like effects in the animal. In any case, the phenotype has been reported previously^{9,10}, and it would seem to be unrelated to the question we are studying, which involves *retention* of DILPs.

Reviewer point 2:

Is insulin downstream of the ecdysone system?

Author response to point 2:

We reported in the original manuscript that hypoxia lowers ecdysone signaling as well as insulin signaling, so, the reviewer asks about the epistatic relationship between these signaling systems. As mentioned in the manuscript, each of these pathways does regulate the other¹¹, but the effect in both directions is inhibitory – an increase in one brings about a decrease in the other. Thus, in our case, in which the activity of *both* signaling systems is suppressed in hypoxia, this cross-regulation appears not to apply. Rather, both effects appear to arise from a common source, namely hypoxia. However, to address the reviewer’s point, we tested explicitly whether ecdysone signaling is involved in the size difference between hypoxic and normoxic animals by supplementing animals under hypoxia with ecdysone in their medium. We found that this treatment leads to a further decrease in size in hypoxia (Fig. S4E), presumably by reducing the larval growth period. Thus, the reduced ecdysone signaling observed in hypoxia does not underlie the accompanying size defects. We thank the reviewer for requesting that we tie up this loose end.

Reviewer point 3:

How do the btl-RNAi pupae die? Not clear as to what hypoxia-induced damage is here? do the authors have any evidence?

Author response to point 3:

We assume that these animals succumb to the effects of tissue hypoxia in this situation because they cannot respond to this stress through tracheal growth, which was the intended “message” of this figure. Whether this is brought about directly by low oxygen levels, through tissue necrosis for example, is not known. It may be an interesting subject for future studies, but we think the precise manner of death in this instance lies outside the scope of this study.

Reviewer point 4:

How do the authors explain that the fat droplets decrease in size and quantity in the fat bodies under hypoxia in *btl*-deficient flies?

Author response to point 4:

We reported in our original submission that *breathless-RNAi* leads to an increase in the size of lipid droplets stored in the fat body. In exploring the underlying mechanism to address the reviewer's question, we found that HIF-1a Prolyl Hydroxylase (Hph) regulates the activity of the Tor pathway in the fat body (Fig. 7A-7D), through a Sima/HIF-1a-independent mechanism, in response to levels of both oxygen and dietary amino acids. The Tor pathway is a master integrator of cell-autonomous growth-regulatory signals, and the placement of Hph upstream of it is of major significance. We now show that the effects of hypoxia on lipid droplets involve Hph and Tor, and that HIF-1a is not involved in this process (Fig. 7C). This finding not only explains the lipid-droplet phenotype observed in the original manuscript, but also provides a mechanistic basis for interaction between amino-acid levels, oxygen availability, Tor activity, and cell-autonomous and systemic growth regulation.

In summary, we have addressed all the specific comments made by the reviewers. Furthermore, we believe that we have significantly further increased the novelty of our work by placing both oxygen and amino-acid availability upstream of Hph function in the fat body, as well as splitting the pathway downstream of Hph into (1) a Sima/HIF-1a-independent branch that regulates Tor activity (and downstream physiology such as lipid metabolism) and (2) a Tor-independent, Sima/HIF-1a-dependent branch that regulates body growth systemically via effects on insulin signaling. We believe that our finding represents conceptual advances solving long-standing questions of how organisms adapt their growth to environmental conditions.

References

- 1 Wong, D. M., Shen, Z., Owyang, K. E. & Martinez-Agosto, J. A. Insulin- and warts-dependent regulation of tracheal plasticity modulates systemic larval growth during hypoxia in *Drosophila melanogaster*. *PLoS One* **9**, e115297, doi:10.1371/journal.pone.0115297 (2014).
- 2 Reiling, J. H. & Hafen, E. The hypoxia-induced paralogs Scylla and Charybdis inhibit growth by down-regulating S6K activity upstream of TSC in *Drosophila*. *Genes Dev* **18**, 2879-2892, doi:10.1101/gad.322704 (2004).
- 3 Brugarolas, J. *et al.* Regulation of mTOR function in response to hypoxia by REDD1 and the TSC1/TSC2 tumor suppressor complex. *Genes Dev* **18**, 2893-2904, doi:10.1101/gad.1256804 (2004).
- 4 Masson, N. & Ratcliffe, P. J. Hypoxia signaling pathways in cancer metabolism: the importance of co-selecting interconnected physiological pathways. *Cancer Metab* **2**, 3, doi:10.1186/2049-3002-2-3 (2014).
- 5 Colombani, J. *et al.* A nutrient sensor mechanism controls *Drosophila* growth. *Cell* **114**, 739-749, doi:10.1016/S0092-8674(03)00713-X (2003).
- 6 Delanoue, R. *et al.* *Drosophila* insulin release is triggered by adipose Stunted ligand to brain Methuselah receptor. *Science* **353**, 1553-1556, doi:10.1126/science.aaf8430 (2016).
- 7 Koyama, T. & Mirth, C. K. Growth-Blocking Peptides As Nutrition-Sensitive Signals for Insulin Secretion and Body Size Regulation. *PLoS Biol* **14**, e1002392, doi:10.1371/journal.pbio.1002392 (2016).
- 8 Rajan, A. & Perrimon, N. *Drosophila* cytokine unpaired 2 regulates physiological homeostasis by remotely controlling insulin secretion. *Cell* **151**, 123-137, doi:10.1016/j.cell.2012.08.019 (2012).
- 9 Sato-Miyata, Y., Muramatsu, K., Funakoshi, M., Tsuda, M. & Aigaki, T. Overexpression of *dilp2* causes nutrient-dependent semi-lethality in *Drosophila*. *Front Physiol* **5**, 147, doi:10.3389/fphys.2014.00147 (2014).
- 10 Honegger, B. *et al.* Imp-L2, a putative homolog of vertebrate IGF-binding protein 7, counteracts insulin signaling in *Drosophila* and is essential for starvation resistance. *J Biol* **7**, 10, doi:10.1186/jbiol72 (2008).
- 11 Colombani, J. *et al.* Antagonistic actions of ecdysone and insulins determine final size in *Drosophila*. *Science* **310**, 667-670, doi:10.1126/science.1119432 (2005).

REVIEWERS' COMMENTS:

Reviewer #1 (Remarks to the Author):

The authors have convincingly addressed all concerns that I had raised in the original review. The revised manuscript is a much improved version of the original submission. The authors have done a wonderful job in highlighting the significance of their original work. Moreover, their new finding that Hph integrates both hypoxia and AA inputs in the FB and affects growth by independently regulating TOR and HIF-1-dependent insulin release adds a significant new dimension to the work that will be of interest to a very broad audience.

The authors have also very satisfactorily addressed all experimental concerns that I had raised in the original review.

Overall I now feel that the current body of work is excellent original research that deserves a reconsideration for Nature Communications.

Only minor edit:

Reviewer 1, Point 4: The suggested references are still missing although they claim to have included them in the manuscript (unless they are there separately in the supplement section). Relevant line in the text with other references missing: Page 20 line 27-29

Reviewer #3 (Remarks to the Author):

This is a revised manuscript. The authors have addressed most of the reviewers' questions/concerns in details.

Minor:

In the legend of Figure S4, line 7: "..... b Effects of 20-hydroxyecdysone....." should be "..... e Effects of 20-hydroxyecdysone....."

Reviewer #1 (Remarks to the Author):

Reviewer: *The authors have convincingly addressed all concerns that I had raised in the original review. The revised manuscript is a much improved version of the original submission. The authors have done a wonderful job in highlighting the significance of their original work. Moreover, their new finding that Hph integrates both hypoxia and AA inputs in the FB and affects growth by independently regulating TOR and HIF-1-dependent insulin release adds a significant new dimension to the work that will be of interest to a very broad audience.*

The authors have also very satisfactorily addressed all experimental concerns that I had raised in the original review.

Response: We are pleased that the reviewer finds that we have addressed all concerns and greatly improved our manuscript. We are also glad that we have both expanded the significance of our work, as well as conveyed this significance more clearly.

Reviewer: *Overall I now feel that the current body of work is excellent original research that deserves a reconsideration for Nature Communications.*

Response: Thank you very much!

Reviewer: *Only minor edit:*

Reviewer 1, Point 4: The suggested references are still missing although they claim to have included them in the manuscript (unless they are there separately in the supplement section). Relevant line in the text with other references missing: Page 20 line 27-29.

Response: We apologize for our omission – we wrote the response letter but forgot to make the actual change. Dawdle now first appears, along with the two citations the reviewer mentioned, in the introduction along with other fat-body-derived factors.

Reviewer: *Minor:*

In the legend of Figure S4, line 7: "..... b Effects of 20-hydroxyecdysone....." should be "..... e Effects of 20-hydroxyecdysone....."

Response: We thank the reviewer for catching the mistake. We have corrected it.

Thus, we have addressed the minor errors caught by our attentive Reviewers, and we thank them for their help. It has been our pleasure to work with them and with the *Nature Communications* Scientific and Editorial teams in producing a much-improved manuscript.

As always, please do not hesitate to contact me if you need further information.